# Gradient strikes back: How filtering out high frequencies improves explanations

## Abstract

Recent years have witnessed an explosion in the development of novel prediction-based attribution methods, which have slowly been supplanting older gradient-based methods to explain the decisions of deep neural networks. However, it is still not clear why prediction-based methods outperform gradient-based ones. Here, we start with an empirical observation: these two approaches yield attribution maps with very different power spectra, with gradient-based methods revealing more high-frequency content than prediction-based methods. This observation raises multiple questions: What is the source of this high-frequency information, and does it truly reflect decisions made by the system? Lastly, why would the absence of high-frequency information in prediction-based methods yield better explainability scores along multiple metrics? We analyze the gradient of three representative visual classification models and observe that it contains noisy information emanating from high-frequencies. Furthermore, our analysis reveals that the operations used in Convolutional Neural Networks (CNNs) for downsampling appear to be a significant source of this high-frequency content – suggesting aliasing as a possible underlying basis. We then apply an optimal low-pass filter for attribution maps and demonstrate that it improves gradient-based attribution methods. We show that *(i)* removing high-frequency noise yields significant improvements in the explainability scores obtained with gradient-based methods across multiple models – leading to *(ii)* a novel ranking of state-of-the-art methods with gradient-based methods at the top. We believe that our results will spur renewed interest in simpler and computationally more efficient gradient-based methods for explainability.

## 1 Introduction

Explaining and interpreting the decision of AI architectures is an important area of research towards enabling the development of more interpretable models. Explainability methods (XAI) aim to provide insights into the strategies used by models to arrive at their decision. This is expected to lead to the development of better models that are more accurate, robust, and better aligned with humans.

One of the first attribution methods proposed, "Saliency" [1], consists of back-propagating a model's decision back to an input image to highlight areas that most affected the final decision. The method remains relatively simple and computationally efficient, but it is also known to be noisy and to lead to attribution maps that are often hard to interpret. Multiple methods have been proposed since to try to improve on these limitations. These methods fall broadly into two main classes. (i) Gradient-based methods extend Saliency [1] by smoothing the resulting attribution maps [2–8]. However, these so-called white-box methods require access to all the model's components, which is not always

---

† The authors contributed equally.

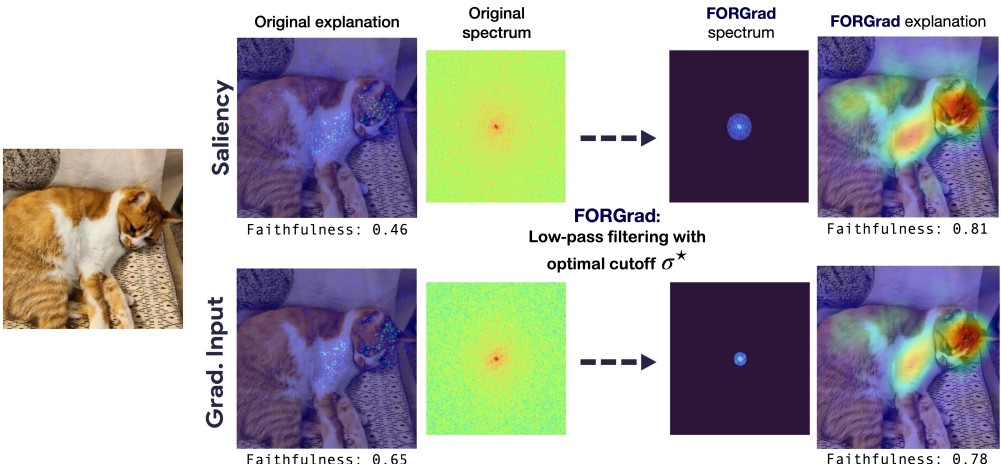

Figure 1: **Effect of FORGrad on gradient-based attribution methods.** We show that for an input image (left), the initial explanations from two gradient-based methods are plagued by noise as indicated by the high power in the high-frequency range of their respective spectra. Filtering the explanations with **FORGrad** yields improved explanations (right).

possible. Conversely, prediction-based methods, also called black-box methods [3, 9–11], alter the input of the model to produce an explanation based on the resulting change in the output. Those methods are computationally inefficient and are known to sometimes fail to capture all the diagnostic information, but they currently lead to the best fidelity scores across all explainability methods. Overall, there are dozens of attribution methods available but relatively little is understood about what makes certain methods more accurate than others.

Here, we start with the observation made across multiple studies [12, 4, 13] that the attribution maps derived with Saliency are very noisy. Generally, these maps highlight sparse pixel activations around a region of interest, and they are often hard to interpret. Because Saliency is simply the gradient of the score function with respect to the input, we suggest that the noise originates from the gradient itself: in other words, because the gradient is noisy, the explanation provided by Saliency is also noisy. To try to better understand the origin of this noise, we compare the Fourier power spectra of gradient-based methods (including Saliency) against prediction-based methods and observe that they differ quite markedly. We discern significant differences between the two classes of approaches, with gradient-based methods returning higher frequency content and prediction-based methods returning lower frequency content. In the remainder of this paper, we will show that:

- The gradient is indeed noisy, and this noise is especially present in the high-frequencies.

- We then look for the origin of these high frequencies in vision models. Our findings show that downsampling operations (via MaxPooling or strides) are the main sources of high frequencies, and training the model does not alleviate the issue.

- We then propose to repair Saliency – as well as other gradient-based methods – by introducing **FORGrad** (FOurier Reparation of the Gradient). This method consists in estimating the optimal amount of high frequencies to remove per model to make gradient-based methods surpass the prediction-based family of attribution methods.

## 2 Related Work

**Attribution methods for black-box models** Various methods have been developed to compute importance scores for individual pixels or groups of pixels. For black-box (prediction-based) attribution methods, the analytical form and potential internal states of the model are unknown. The first method, Occlusion [3], masks individual image regions, one at a time, using an occluding mask set to a baseline value. The corresponding prediction scores are assigned to all pixels within the occluded region, providing an easily interpretable explanation. However, occlusion fails to account for the joint (higher-order) interactions between multiple image regions. For instance, occluding two image regions individually may only have a minimal impact on the model's prediction, such as removing a single eye or mouth component from a face. However, occluding these two regions together may lead

to a substantial change in the model's prediction if these regions interact non-linearly, as expected in a deep neural network. Sobol [10], along with related methods such as LIME [11] and RISE [9], address this problem by randomly perturbing multiple regions of the input image simultaneously. Interestingly, recent studies, including RISE [9] and Sobol [10], have demonstrated that black-box attribution methods can rival and even surpass the commonly used white-box methods without relying on internal states.

**Attribution methods for white-box models**   The gradient-based methods, that we propose to improve here, were first introduced in [14] and improved in [2–4]. They consist in explaining the decisions of a model by back-propagating the gradient from the output to the input, indicating which pixels affect the decision score the most. However, this family of methods is limited because they focus on the influence of individual pixels in an infinitesimal neighborhood in the input image. For instance, it has been shown that gradients often vanish when the prediction score to be explained is near the maximum value [6]. Integrated Gradient [6] and SmoothGrad [5] partially address this issue by accumulating gradients. Another family of attribution methods relies on the neural network's activation, like CAM [7], which computes an attribution score based on a weighted sum of feature channel activities – right before the classification layer. GradCAM [8] extends CAM via the use of gradients, re-weighting each feature channel to take into account their importance for the predicted class. Nevertheless, the choice of the layer has a huge impact on the quality of the explanation. Our contribution proposes to overcome some of the mentioned issues by removing the noise present in the gradients in the form of high frequencies.

**Fourier analysis of vision models**   Very little work has been proposed to analyze vision models and methods from a Fourier perspective. The closest, [15], used Fourier analysis to investigate the impact of DNNs optimization parameters and methods without a specific focus on vision. Additional work has focused on the analysis and development of adversarial attacks in the Fourier domain, [16, 17], while others [18–20] have proposed to defend against adversarial attacks by transforming the input image in the Fourier domain. Jo and Bengio [21] examined whether CNNs rely on high-level features by using Fourier-filtered images. None of the mentioned studies make a link between explainability and attribution methods with Fourier analysis.

## 3   Decomposing the gradient: An analysis of frequency content in attribution methods

**Notations**   We consider a general supervised learning setting, where a classifier $\boldsymbol{f} : \mathcal{X} \to \mathcal{Y}$ maps images from an input space $\mathcal{X} \subseteq \mathbb{R}^{W \times H}$ to an output space $\mathcal{Y} \subseteq \mathbb{R}$. Let $(\boldsymbol{x}_1, \ldots, \boldsymbol{x}_N)$ be a set of images which contains $N$ samples drawn from a probability distribution $\forall i \in \{1 \cdots n\}, \boldsymbol{x}_i \sim \mathcal{D}$. Moreover, we respectively denote $\mathcal{F}$ and $\mathcal{F}^{-1}$ the Discrete Fourier Transform (DFT) on $\mathbb{R}^{W \times H}$ and its inverse. Therefore: $\forall \boldsymbol{x} \in \mathcal{X}$, $\mathcal{F}(\boldsymbol{x}) \in \mathbb{C}^{W \times H}$ and $(\mathcal{F}^{-1} \circ \mathcal{F})(\boldsymbol{x}) = \boldsymbol{x}$. Additionally, when we visualize the Fourier spectrum, we always shift the low-frequency components to the center of the spectrum. We recall that an attribution method is a function $\boldsymbol{\varphi} : \mathcal{X} \to \mathbb{R}^{W \times H}$ that maps an input of interest to its corresponding importance scores $\boldsymbol{\varphi}(\boldsymbol{x})$. Finally, we denote by $\boldsymbol{\varphi}_\sigma(\boldsymbol{x})$ the attribution method where high frequencies have been filtered using a cutoff value of $\sigma$.

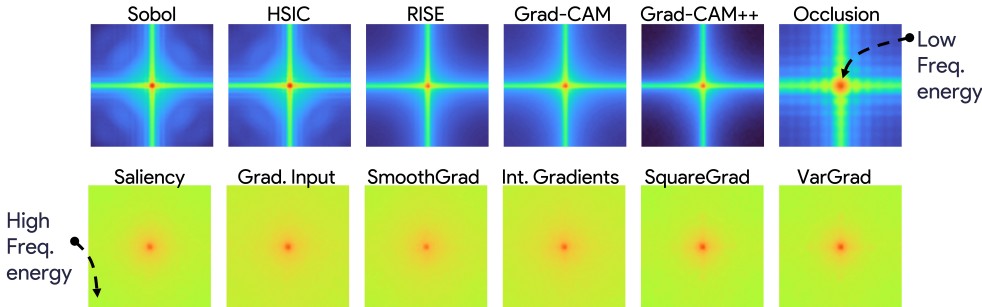

Figure 2: **Fourier footprint of attribution methods.** We show on the top row the Fourier spectrum of prediction-based attribution methods and of the gradient-based methods on the bottom row, computed with a ResNet50. The two families can be distinguished by methods but also by their signature in the Fourier domain. The former method has magnitudes largely concentrated in the low frequencies, while the latter is more spread out: it features non-trivial magnitudes almost everywhere, including in high frequencies.

## 3.1 Different signatures for different categories of methods

In this work, we analyze the Fourier signature of several attribution methods. To do so, we compute the feature map $\varphi(x)$ for most existing attribution methods on representative models of the literature (ResNet50 in Figure 2). From these importance maps, we extract the corresponding amplitude of the Fourier spectrum, $|(\mathcal{F} \circ \varphi)(x)|$. In Figure 2, we show the average power spectra, over 1,000 images, for an array of methods. Upon visual inspection, it is obvious that certain methods tend to emphasize higher frequencies in their explanations, while others concentrate on lower frequencies. Interestingly, these differences can be traced to the class of methods: Black-box methods, which do not rely on gradients, exhibit frequency footprints dominated by very low frequencies, whereas white-box methods exhibit footprints that extend into higher frequencies. To quantify our observations, we employ two metrics to measure the complexity of the attribution maps. The first metric employs a Laplacian-based operator [22, 23] that evaluates the presence of high frequencies in images by analyzing their second derivative. The second metric involves measuring the file size of the image after undergoing lossless compression [24, 25], which we refer to as "High-frequency content" throughout this study (as it can be seen as a loose approximation of Kolmogorov complexity).

Both metrics validate our visual observations, as depicted in Figure 3 (see Laplace quantity in appendix). It is evident that black-box methods (shown in dark in the figure) exhibit fewer high frequencies compared to white-box methods. This observation provides valuable insight into where these methods extract information from the model to compute their explanations.

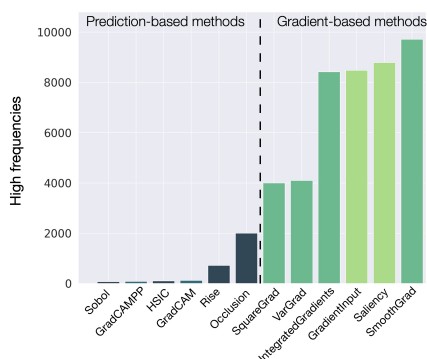

## 3.2 High-frequencies are just noise in the gradient

Naturally, gradient-based methods will be subject to the characteristics of the gradient itself. Consequently, when the gradient is subject to noise, the resulting explanation provided by such methods becomes similarly noisy. In light of this observation, we propose to demonstrate that the gradients obtained from three state-of-the-art models (ResNet50 [26], ViT [27], and ConvNeXT [28]) do indeed contain noise, predominantly present in high-frequency components. To achieve this, we suggest an approach

Figure 3: **High-frequency power in attribution methods.** High-frequency power present in the importance maps derived from different attribution methods. Prediction-based methods produce less high-frequency content than gradient-based methods.

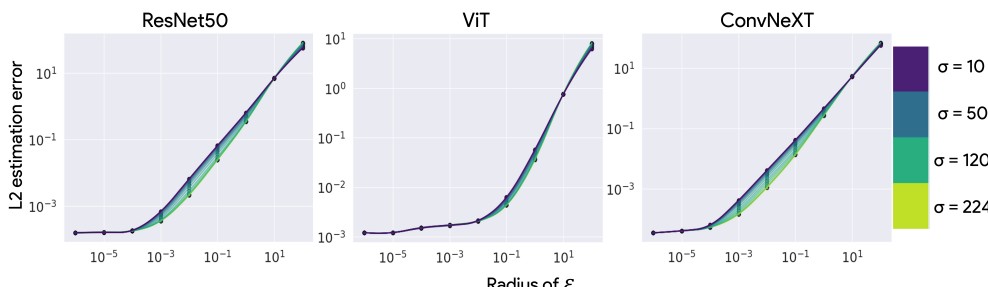

Figure 4: **Evidence for noise in the gradient.** We plot the residual of the first-order approximation of the model, that is $f(x + \varepsilon) \approx f(x) + \varepsilon \nabla_x f(x)$, with the gradient $\nabla f$ filtered at different bandwidths $\sigma$. We sample 100 values of $\varepsilon$ uniformly on $\mathbb{S}_{d-1}$ scaled by the radius, for 1,000 images of the validation set of ImageNet. If high-frequencies contained information necessary for a good linearization of the model then we would observe a gap between the curves of $\sigma = 224$ - where no filter is applied, vs. the curves where we apply a filter - $\sigma < 224$.

that involves selectively removing high-frequency gradient information by employing various frequency cutoffs $\sigma$. By computing residuals between $f(x + \varepsilon)$ and its first order filtered by $\sigma$ decomposition around $x$ that we denote by $f(x) + \varepsilon \nabla_x^\sigma f(x)$, we generate corresponding curves for

different scales of $\varepsilon$, which are presented in Figure 4. As anticipated, our observations reveal that the curves exhibiting reduced high-frequency content (from $\sigma < 224$ to $\sigma = 10$) closely align with the one of the non-filtered gradient ($\sigma = 224$). In other words, the gradient remains approximately as informative, even after removing high-frequency information. This implies that the high-frequency content primarily contains noisy information within the gradient.

### 3.3 Investigating the mechanisms introducing noise

Next, we investigated the underlying operations responsible for the introduction of such content by computing the power of high-frequency content in the gradients at the level of all the layers. Notably, we observed a consistent trend in CNNs where high-frequency content tends to increase and jump at each block, indicative of downsampling operation through strided convolutions or pooling. This observation aligns with the findings of [29, 30], suggesting that downsampling operations via MaxPooling or strided convolution can introduce noise. To verify this hypothesis, we substituted these specific operations in two representative CNNs, namely ResNet50 [26] (which incorporates strided convolutions and MaxPooling) and VGG16 [31] (details in appendix) with AveragePooling. This replacement ensured the preservation of information continuity in the gradient. The resulting plots for ResNet50 (VGG) are presented in Figures 5 - bottom curve (see appendix for VGG), displaying the power of high-frequency content using Kolmogorov image compression and Laplace-operator (see appendix) at each step. The depicted red shades represent the amount of high-frequency content in both models. We observe that prior to the initial dimension change, the quantity of high-frequency content remain comparable, suggesting that operations within a block of the same dimension does not significantly increase the power of high-frequency content. However, with the introduction of a downsampling layer, the curves for each model diverge, indicating a bigger contribution to the introduction of high frequencies by striding or MaxPooling compared to AveragePooling. Our findings corroborate the observations of [29], as the gradients (even averaged) following MaxPooling or strides exhibit checkerboard patterns, providing a plausible explanation for our quantitative observation of increased high-frequency content.
We employ the same pipeline to calculate the high-frequency content for both a random model and a trained model, using the identical set of models including ViT [27]. The resulting curves are depicted in Figures 5 - top curve, for ResNet50 (see appendix for VGG16 and for ViT), showcasing that there is no discrepancy in high-frequency content between the trained and random CNNs. Given our previous section's demonstration that high-frequencies carry negligible information for the model, one would expect that training could potentially eliminate this content, leaving only relevant information to be processed. However, as our observations indicate the absence of such behavior despite the models accomplishing the task, we propose that the models were unable to adapt the gradient's content, thereby suggesting it to be an inherent by-product of downsampling operations. In the case of the ViT, however, training appears to introduce some high frequencies from the initial operation, potentially arising from transformers' pre-processing functions, such as image flattening via patches. These multiple findings suggest that high-frequency content emerges as a by-product of particular operations, predominantly observed in CNNs, which the models are unable to modulate during training. We therefore propose to consider most of the high-frequency content as noise. Consequently, when generating explanations for the models' decisions, it is justifiable to disregard high frequencies as they offer limited or negligible information.

### 3.4 FORGrad: a simple strategy to remove noise

**An adapted $\sigma^\star$ per model**  With **FORGrad**, we propose to remove high-frequency content, considered as noise, in order to obtain an optimal explanation related to the optimal frequency band from the gradient. We therefore propose to apply a low-pass filter on the Fourier spectrum of the gradient, employing multiple frequency cutoffs spaced evenly apart. For each filtered explanation, we compute the score from two different metrics. The first one Deletion – denoted $D(\varphi(\boldsymbol{x})$ [9] – is a measure of the decrease in the likelihood of a particular class as the important pixels (identified by the saliency map) are systematically removed from the image. If the likelihood of the class experiences a rapid decrease, resulting in a small area under the probability curve, this is a strong indication of a good explanation. Complementary, Insertion, $I(\varphi(\boldsymbol{x}))$ [9] measures the significance of the pixels based on their capacity to create an image, and is calculated by measuring the increase in the probability of the class of interest as pixels are added in accordance with the generated importance map. Overall, we propose a heuristic to optimize our $\sigma^\star$, representing the ideal bandwidth maximizing the difference $\sigma^\star = \arg\max_\sigma \mathbb{E}_{\boldsymbol{x}} D(\varphi_\sigma(\boldsymbol{x})) - I(\varphi_\sigma(\boldsymbol{x}))$, combining the score of both metrics, on a subset of the validation set of ImageNet (1,000 images). Using both deletion and

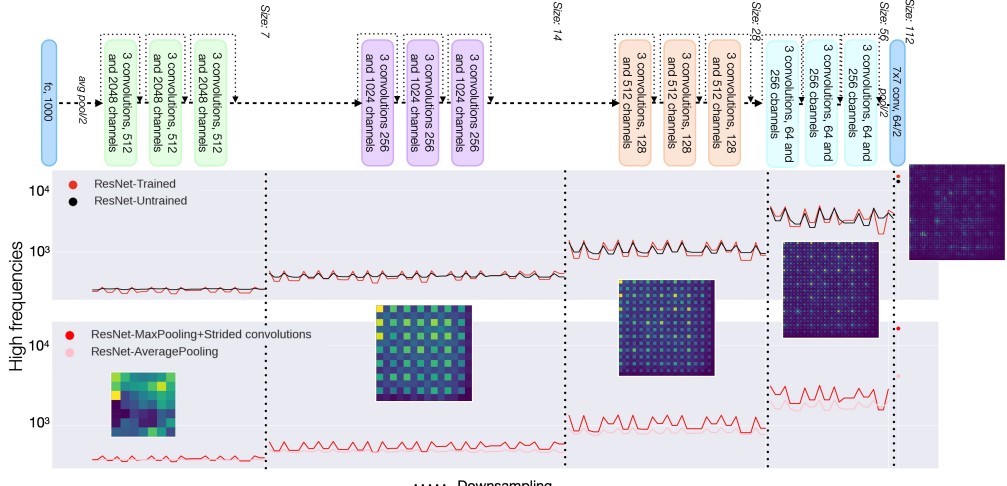

Figure 5: **Evolution of the high-frequency content in Resnet50.** We compute the high-frequency content along the depth of a ResNet50 varying either the weights or the pooling. The top curve represents the trained model, indicated by the red curve, while the untrained model is represented by the black curve. The bottom curve illustrates the impact of different poolings, with MaxPooling and stride shown in dark red and AveragePooling in pink. Each point on the graph corresponds to a layer within the models. In addition, we present visual examples of averaged gradients across 128 images after applying MaxPooling. Despite the averaging process, these examples exhibit checkerboard patterns, serving as a visual demonstration of the presence of high-frequencies.

insertion metrics can provide a more comprehensive evaluation of the quality of the attribution map or saliency map generated for a given model. The deletion metric is useful for identifying important regions of an image that contribute to a model's decision, while the insertion metric is valuable for assessing the quality of the generated saliency map in terms of its ability to reconstruct the original image. By combining both metrics, we aim to assess the quality of the explanations generated by considering the impact of pixel removal and addition on the likelihood and significance of the target class, respectively. In the latter sections, we will consider the faithfulness metric to be the combination [Deletion-Insertion] Additionally, we also evaluate **FORGrad** on a third metric, MuFidelity, $F(\varphi(\boldsymbol{x}))$, [32]. The fidelity correlation metric serves to verify the correlation between the attribution score and a random subset of pixels. To achieve this, a set of pixels is randomly chosen and set to a baseline state, after which a prediction score is obtained. The fidelity correlation metric evaluates the correlation between the decrease in the score and the significance of the explanation for each random subset created.

**Theoretical foundations**    In this section, we build on the empirically demonstrated assumption that the gradient is noisy and prove, through a Fourier perspective, that **FORGrad** effectively recovers the true gradient. Moreover, assuming that the noise is originally Gaussian, we characterize the distribution of the noise in Fourier space. Finally, we propose a convergence bound for SmoothGrad, valid on finite samples, showing that it also recovers the true gradient at the cost of multiple samplings.

We denote $\|\cdot\|_F$ as the Frobenius norm and $\|\cdot\|_2$ as the spectral norm. Note that $\|\cdot\|_2 \leq \|\cdot\|_F$ in order to interpret our results. Finally, we define $\boldsymbol{K}^\sigma \in \{0,1\}^{W \times H}$ as the binary Fourier mask, parameterized by $\sigma$, that we used to filter high frequency, where each element $\boldsymbol{K}^\sigma_{(i,j)}$ is determined as $\bar{\boldsymbol{K}}^\sigma_{(i,j)} = \mathbb{1}_{|i-\frac{W}{2}| \leq \sigma} \mathbb{1}_{|j-\frac{H}{2}| \leq \sigma}$, with $\mathbb{1}$ the indicator function abd $\bar{\boldsymbol{K}}^\sigma = 1 - \boldsymbol{K}^\sigma$. As we have discussed above, the gradient of deep models is noisy, and in the following work, we consider that we only have access to $\nabla_{\boldsymbol{x}} \hat{f}(\boldsymbol{x})$, a noisy estimator of $\nabla_{\boldsymbol{x}} f(\boldsymbol{x})$ such that $\nabla_{\boldsymbol{x}} \hat{f}(\boldsymbol{x}) = \nabla_{\boldsymbol{x}} f(\boldsymbol{x}) + \varepsilon$ with $\varepsilon \in \mathbb{R}^{W \times H}$. We do not assume any randomness for the noise so far. The following proposition develops the squared residual of the filtered noisy gradient as compared to the true one. Under the condition of finding the optimal filter, the gap between the two is naturally norm of the remaining noise post filtering.

**Proposition 3.1.** *Let $\boldsymbol{f} : \mathcal{X} \to \mathcal{Y}$ a predictor, and denote $\nabla \hat{\boldsymbol{f}} = \nabla \boldsymbol{f} + \varepsilon$ as the noisy gradient of $\boldsymbol{f}$, with $\varepsilon \in \mathbb{R}^{W \times H}$. For $\sigma^* = \inf \left\{ \sigma : \|\mathcal{F}(\nabla \boldsymbol{f}) \odot \bar{\boldsymbol{K}}^\sigma\|_F^2 = 0 \right\}$, we have*

$$\|\mathcal{F}^{-1}(\mathcal{F}(\nabla \hat{\boldsymbol{f}}) \odot \boldsymbol{K}^{\sigma^*}) - \nabla \boldsymbol{f}\|_F^2 = \|\mathcal{F}^{-1}(\mathcal{F}(\varepsilon) \odot \boldsymbol{K}^{\sigma^*})\|_F^2 \leq \|\varepsilon\|_F^2, \tag{1}$$

where $\odot$ is the Hadamard product, $\boldsymbol{K}^{\sigma^*}$ a binary mask for low-pass filtering of frequency $\sigma$, and $\bar{\boldsymbol{K}}^{\sigma^*}$ is the opposite mask.

*Remark* 3.2. This result holds as long as we find $\sigma^*$. There always exists a $\sigma^*$ as the set always contains $\sigma = \max(H, W)$ which does not alter the Fourier spectrum of an image of size $W \times H$. However, finding $\sigma^*$ poses a challenge, leading us to leverage XAI metrics as a heuristic.

With the information that the remaining gap between the filtered estimator and the true gradient is the remaining noise, of which the norm is upper bounded by the one of the original noise, we aim at measuring the reduction of the noise. In that way, we demonstrate the always-positive effect of **FORGrad** on gradient methods. In particular, under the assumption of Gaussian noise, we derive the distribution of the ratio $\|\boldsymbol{\varepsilon}\|_F^2 / \|\mathcal{F}^{-1}(\mathcal{F}(\boldsymbol{\varepsilon}) \odot \boldsymbol{K}^{\sigma^*})\|_F^2$.

**Proposition 3.3.** *Let the noise $\boldsymbol{\varepsilon} \in \mathbb{R}^{W \times H}$ follow a normal distribution $\boldsymbol{\varepsilon} \sim \mathcal{N}(0, \varsigma)^{\otimes N}$. Then the norm of the Fourier spectra of the noise $\|\mathcal{F}(\boldsymbol{\varepsilon})\|_F^2 \sim \Gamma(k = 2WH, \theta = \varsigma^2 WH)$ and filtered noise $\|\mathcal{F}(\boldsymbol{\varepsilon}) \odot \boldsymbol{K}^{\sigma}\|_F^2 \sim \Gamma(k = 8\sigma^2, \theta = 4\varsigma^2 \sigma^2)$ follow Gamma distributions.*

*Therefore, the ratio of the two distributions $R = \|\mathcal{F}(\boldsymbol{\varepsilon})\|_F^2 / \|\mathcal{F}(\boldsymbol{\varepsilon}) \odot \boldsymbol{K}^{\sigma}\|_F^2$ follows a Beta prime distribution $R \sim \beta'\left(2N, 8\sigma^2, 1, \frac{WH}{4\sigma^2}\right)$.*

This result allows us to directly compute the distribution of the ratio of the norm of the original noise on the norm of the filtered noise (up to a scaling factor, by Parseval's Theorem). Naturally, this distribution depends on the parameter $\sigma$ of the filtering. From this, we can deduce probabilistic results, such as, for $\sigma = 10$ and $\varsigma = 1$, the ratio of the norms is larger than 70 with probability almost one.

Finally, in the following proposition, we obtain a non-asymptotic result on the concentration of the SmoothGrad procedure to its expected value based on the Matrix Bernstein inequality [33].

**Proposition 3.4.** *We recall that SmoothGrad is defined as $SG = \frac{1}{n} \sum_{i=1}^{n} \nabla_{\boldsymbol{x}} \boldsymbol{f}(\boldsymbol{x} + \delta_i)$ with $\forall_{i=1,\ldots,n} \delta_i \in \mathcal{N}(0, \varsigma)$. Here we compute SG on the noisy estimate of the gradient $\nabla_{\boldsymbol{x}} \hat{\boldsymbol{f}}(\boldsymbol{x} + \delta_i) \in \mathbb{R}^{W \times H}$, which is then a random matrix $\widehat{SG}$. Assuming our predictor $\boldsymbol{f} \in L\text{-}Lip(\mathcal{X})$ is $L$-Lipschitz. We denote $\|\cdot\|_2$ as the spectral norm, and define the variance as $\mathbb{V}(SG) = \max\left(\|\mathbb{E}((SG - \mathbb{E}SG) \cdot (SG - \mathbb{E}SG)^T)\|_2, \|\mathbb{E}((SG - \mathbb{E}SG)^T \cdot (SG - \mathbb{E}SG))\|_2\right)$. We then have, for $t > 0$,*

$$\mathbb{P}\left(\|\widehat{SG} - \mathbb{E}SG\|_2 \geq t\right) \leq (W + H) \cdot \exp\left(\frac{-t^2 n^2 / 2}{\mathbb{V}(\widehat{SG}) + 2Lt/3}\right). \tag{2}$$

Our results suggest that in order to effectively eliminate noise using the SmoothGrad method, several iterations are required as opposed to ours. For instance, to be at least $t = \frac{L}{10}$ away from its expected value, with probability at most 0.01, we need $n \approx 700$ iterations, for $\varsigma = 1$. Furthermore, the noisy SmoothGrad gradually approaches the expected outcome of the non-noisy SmoothGrad. Additionally, SmoothGrad alleviate the noise but at the cost of employing Monte-Carlo sampling.

## 4 Gradient-based methods perform better and are more efficient

**The new explanations are free from noise** Figure 6 presents qualitative examples of corrected gradients obtained using the Gradient Input method [34] combined with **FORGrad**. As we analyze the different images, we observe that gradually removing high frequencies from the gradients has a notable impact on the resulting explanation. The initially noisy patterns transform into larger patches until the saliency map effectively highlights the key features that represent the object for categorization. However, it is crucial to consider the optimal value of $\sigma^{\star}$, as exceeding this threshold leads to the map spreading too widely and the explanation becoming less informative. This observation is further supported by the curve on the right, which demonstrates the evolution of the faithfulness score as $\sigma$ changes. Prior to finding the optimal $\sigma$, the faithfulness score fluctuates around the initial value before gradually increasing to reach its optimal level. As expected, when all the information is removed (represented by the last point on the x-axis), the fidelity score drops to zero.

**A new ranking of attribution methods** We apply **FORGrad** on all the gradient-based attribution methods and report the scores in Table 1 for three models: ResNet50 [26], ViT [27] and ConvNeXT [28]. GradCAM methods can't be tested on ViT because they are based on convolution so are limited to CNNs. We can observe two notable findings. Firstly, it is rare to encounter cases where

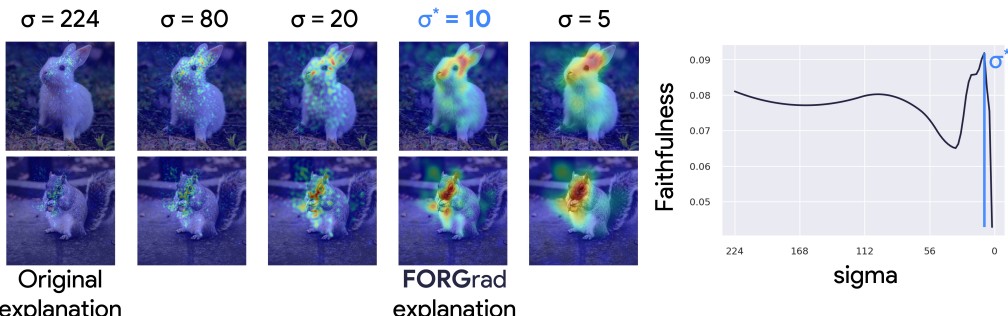

σ = 224    σ = 80    σ = 20    σ* = 10    σ = 5

Original explanation          FORGrad explanation

Figure 6: **FORGrad : selection of the optimal** $\sigma$. With **FORGrad**, we aim to derive the explanation from the gradient's corrected version. To achieve this, we determine the optimal cutoff of high frequencies in order to maximize the faithfulness of the explanation. The images displayed on the left illustrate the progression for different cutoff values. On the right side, the curve represents the variation of the metric across 1,000 images from the validation set of ImageNet, as a function of $\sigma$, representing the cutoff value. $\sigma^{\star}$ represents the optimal value, selected as the one maximizing the faithfulness score, equivalent to [Deletion-Insertion].

| | | ResNet50 | | | | ViT | | | | ConvNeXT | | | |
|---|---|---|---|---|---|---|---|---|---|---|---|---|---|
| | | Del.(↓) | Ins.(↑) | Fid.(↑) | Comp. | Del.(↓) | Ins.(↑) | Fid.(↑) | Comp. | Del.(↓) | Ins.(↑) | Fid.(↑) | Comp. |
| Gradient-based | Saliency[1] | 0.77 | 0.85 | 0.07 | $\Theta(2)$ | 0.77 | 0.82 | 0.01 | $\Theta(2)$ | 0.85 | 0.86 | 0.05 | $\Theta(2)$ |
| | Saliency⋆ | 0.74 | 0.90 | 0.15 | $\Theta(2)$ | 0.81 | 0.89 | 0.03 | $\Theta(2)$ | 0.82 | 0.88 | 0.06 | $\Theta(2)$ |
| | GradInput[34] | 0.76 | 0.87 | 0.05 | $\Theta(2)$ | 0.78 | 0.88 | 0.01 | $\Theta(2)$ | 0.83 | 0.89 | 0.05 | $\Theta(2)$ |
| | GradInput⋆ | 0.74 | 0.88 | 0.13 | $\Theta(2)$ | 0.74 | 0.89 | **0.05** | $\Theta(2)$ | 0.81 | 0.88 | 0.07 | $\Theta(2)$ |
| | SmoothGrad[5] | 0.74 | 0.89 | 0.08 | $\Theta(200)$ | 0.80 | 0.87 | 0.03 | $\Theta(200)$ | 0.86 | 0.86 | 0.05 | $\Theta(200)$ |
| | SmoothGrad⋆ | **0.72** | **0.93** | **0.19** | $\Theta(200)$ | 0.78 | **0.92** | 0.04 | $\Theta(200)$ | 0.82 | 0.88 | 0.05 | $\Theta(200)$ |
| | VarGrad[35] | 0.74 | 0.91 | 0.07 | $\Theta(200)$ | **0.72** | 0.88 | 0.01 | $\Theta(200)$ | 0.89 | 0.86 | 0.02 | $\Theta(200)$ |
| | VarGrad⋆ | 0.73 | 0.91 | 0.18 | $\Theta(200)$ | 0.74 | 0.90 | 0.02 | $\Theta(200)$ | **0.80** | 0.88 | 0.02 | $\Theta(200)$ |
| | Int.Grad[6] | 0.75 | 0.88 | 0.06 | $\Theta(200)$ | 0.78 | 0.86 | 0.01 | $\Theta(200)$ | 0.82 | 0.90 | 0.05 | $\Theta(200)$ |
| | Int.Grad⋆ | 0.74 | 0.89 | 0.15 | $\Theta(200)$ | 0.79 | 0.87 | 0.03 | $\Theta(200)$ | 0.81 | 0.90 | 0.05 | $\Theta(200)$ |
| Prediction-based | GradCAM[8] | 0.78 | 0.92 | 0.06 | $\Theta(2)$ | n.a | n.a | n.a | n.a | 0.87 | 0.92 | 0.06 | $\Theta(2)$ |
| | GradCAM++[36] | 0.75 | **0.93** | 0.08 | $\Theta(2)$ | n.a | n.a | n.a | n.a | 0.90 | 0.92 | 0.02 | $\Theta(2)$ |
| | Occlusion[34] | 0.75 | 0.85 | 0.06 | $\Theta(1024)$ | 0.79 | 0.83 | 0.01 | $\Theta(1024)$ | 0.83 | 0.88 | 0.07 | $\Theta(1024)$ |
| | HSIC[37] | **0.72** | 0.92 | 0.05 | $\Theta(2000)$ | 0.77 | 0.91 | 0.02 | $\Theta(2000)$ | **0.80** | 0.92 | 0.05 | $\Theta(2000)$ |
| | Sobol[38] | 0.74 | 0.92 | 0.06 | $\Theta(4000)$ | 0.79 | 0.91 | 0.02 | $\Theta(4000)$ | 0.82 | 0.93 | **0.08** | $\Theta(4000)$ |
| | RISE[9] | 0.76 | **0.93** | 0.07 | $\Theta(8000)$ | 0.80 | **0.92** | 0.01 | $\Theta(8000)$ | 0.84 | **0.94** | 0.07 | $\Theta(8000)$ |

Table 1: **Results on Faithfulness metrics**. Deletion, Insertion, and Fidelity scores obtained on 1,000 ImageNet validation set images, on an Nvidia V100 (For Deletion, lower is better and for Insertion and Fidelity, higher is better). Complexity $\Theta$ (Comp.) corresponds to the number of forward + backward passes required for computation, up to a factor that depends on the model. The first and second best results are in **bold** and underlined.

the scores after applying **FORGrad** are lower than the scores obtained before. In such instances, the decrease in scores is typically observed in only one metric, either Deletion or Insertion. However, since the other metric is optimized, the overall Faithfulness, as measured by [Deletion-Insertion], remains at least as good as before. Secondly, even without explicitly optimizing the Fidelity metric, we observe an improvement in this score across all methods and the three models analyzed. Furthermore, after applying **FORGrad**, we observe that the scores of several gradient-based methods surpass or at least match those of prediction-based methods. Notably, these gradient-based methods offer the additional advantage of being significantly more computationally efficient, as evident from the complexity column. In order to determine the best method for each model, we propose to aggregate the scores from the three metrics to obtain a single global score for each method and model. This resulting score, is denoted as $I(\varphi(\boldsymbol{x})) + F(\varphi(\boldsymbol{x})) - D(\varphi(\boldsymbol{x}))$, corresponding to the sum of 1-Deletion, Insertion and Fidelity score. Interestingly, in Table 2, we observe that the rankings change when we incorporate **FORGrad** into the analysis. This shift leads to the inclusion of at least two gradient-based methods among the top-5 for all three models. In the case of ResNet50, all five of the top-performing methods are gradient-based, whereas only one of them occupied a position in the previous ranking. Although some prediction-based methods, such as *Sobol* and *HSIC*, consistently

| | ResNet50 | | ViT | | ConvNeXT | |
|---|---|---|---|---|---|---|
| | Original | **FORGrad** | Original | **FORGrad** | Original | **FORGrad** |
| 1 | GradCAM++ | **SmoothGrad**⋆ | VarGrad | **SmoothGrad**⋆ | Sobol | Sobol |
| 2 | HSIC | **VarGrad**⋆ | HSIC | **VarGrad**⋆ | RISE | RISE |
| 3 | RISE | **Saliency**⋆ | Sobol | HSIC | HSIC | HSIC |
| 4 | Sobol | **Int.Grad**⋆ | RISE | Sobol | Occlusion | **GradInput**⋆ |
| 5 | VarGrad | **GradInput**⋆ | GradInput | RISE | GradCAM | **Int.Grad**⋆ |

Table 2: **Global ranking before (original) and after FORGrad.** For each model, we show the 5 attribution methods with highest metrics, before and after applying **FORGrad**. The explanation maps were computed on 1000 images from the validation set of ImageNet, based on an aggregation of the three metrics computed by $I(\varphi(\boldsymbol{x})) + F(\varphi(\boldsymbol{x})) - D(\varphi(\boldsymbol{x}))$.

demonstrate good performance, we demonstrate that gradient-based methods such as *SmoothGrad* and *VarGrad* now perform nearly as well, with the added advantage of computational efficiency.

## 5   Limitations

In our study, we have proposed to find an optimal $\sigma$ value representing an ideal cutoff to improve explanations of gradient-based methods. However, we acknowledge that this optimal value is highly dependent on the dataset, perhaps more so than on the model itself. Furthermore, while we have chosen a single value that maximizes the scores across 1,000 images, it may be beneficial to use different values for individual images, but would increase the computational costs. We also optimize our value of $\sigma$ only on 2 metrics, deletion and insertion. Even though it turns out to also increase the fidelity score, we could potentially obtain even better results by optimizing on this metric as well. It's however, once again, a very resource-consuming method that we chose to avoid. Furthermore, in our ranking computation, we combine metrics that do not precisely capture the same information. While Deletion and Insertion can be aggregated, particularly since we optimize the difference between them, it should be noted that Deletion, Insertion, and Fidelity are not directly comparable even if they range between 0 and 1. We have proposed one approach to integrate these metrics and derive a ranking based on the three scores. However, an alternative could involve producing separate rankings for each individual score. If we had followed this approach, the **FORGrad** methods would have emerged as the top-5 for both ResNet50 and ViT, according to MuFidelity.

## 6   Conclusion

This work started with an empirical observation: prediction-based and gradient-based methods exhibit distinct power spectra in their attribution maps – with gradient-based methods exhibiting higher power in the high frequencies compared to prediction-based methods. This led us to wonder whether the frequency content of model gradients is merely noisy information. We demonstrate that removing this content does not impair our ability to approximate the gradient and conclude that high frequencies predominantly carry non-essential information. We further conducted an in-depth analysis of gradient frequency content in CNNs across processing layers and found that downsampling operations, such as max pooling and striding, contribute to the introduction of high frequencies. This points to model aliasing as a likely cause of this high-frequency content. Interestingly, even with training, CNNs are unable to prevent this phenomenon. These results hence raise the question: Could high-frequencies be filtered out to improve the explanations derived from attribution methods? We design an optimal filter, $\sigma^{\star}$, and show that the filtering of attribution maps leads to significant improvements in the quality of the explanations. These improvements were most pronounced for gradient-based methods, which ended up approaching and sometimes even surpassing the much more compute-intensive prediction-based methods. Overall, our work leads to a surprising result – that the almost forgotten gradient-based methods turn out to contain all the information needed to provide a faithful explanation of a model's decision and that they can be as interpretable as the newest methods. In future work, it would be worth exploring the influence of this noise on the model's performance and evaluating whether replacing certain operations that introduce noise could affect both the accuracy and robustness of the models. Furthermore, considering that many adversarial attacks are gradient-based and often exploit additive noise patterns, it is worth investigating whether these attacks target the noisy high-frequency content in the gradients and whether they might be prevented by using operations not introducing high-frequencies.

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
