# Supplementary material:
## Gradient strikes back: How filtering out high frequencies improves explanations

## 1 A Proofs

In this section, we build on the assumption – that we empirically showed in the paper – that the gradient is noisy and demonstrate, through a Fourier perspective, that **FORGrad** method effectively recovers the true gradient. We then propose a convergence bound for SmoothGrad, showing that it also recovers the true gradient at the cost of multiple samplings, and provide convergence bounds as well.

As a recall, we still consider a predictor $\boldsymbol{f}(\cdot)$ that maps datum from an input space $\mathcal{X} \subseteq \mathbb{R}^{W \times H}$ (with $W, H$ being positive integers) to an output space $\mathcal{Y} \subseteq \mathbb{R}$. Moreover, $\mathcal{F}$ and $\mathcal{F}^{-1}$ still denote the Discrete Fourier Transform (DFT) on $\mathcal{X}$ and its inverse. We assume that $\boldsymbol{f}$ is $L$-lipschitz. We recall that a function $\boldsymbol{f}$ is said $L$-lipschitz $\boldsymbol{f} \in Lip(\mathcal{X})$ if and only if $\forall (\boldsymbol{x}, \boldsymbol{z}) \in \mathcal{X}^2, ||\boldsymbol{f}(\boldsymbol{x}) - \boldsymbol{f}(\boldsymbol{z})|| \leq L||\boldsymbol{x} - \boldsymbol{z}||$, with $L \in \mathbb{R}$. Finally, we define $\boldsymbol{K}^\sigma \in \{0, 1\}^{W \times H}$ as the binary mask parametrized by $\sigma$ that we used to filter high frequency, where each element $\boldsymbol{K}^\sigma_{(i,j)}$ is determined as follows:

$$\boldsymbol{K}^\sigma_{(i,j)} = \begin{cases} 1, & \text{if } |i - \frac{W}{2}| \leq \sigma \text{ and } |j - \frac{H}{2}| \leq \sigma, \\ 0, & \text{otherwise.} \end{cases}$$

**Definition A.1.** *Noisy gradient. Let $\boldsymbol{f}$ be differentiable and $\nabla_{\boldsymbol{x}} \boldsymbol{f}(\boldsymbol{x})$ denote the gradient of $\boldsymbol{f}$ in $\boldsymbol{x}$. We consider that we only have access to $\nabla_{\boldsymbol{x}} \hat{\boldsymbol{f}}(\boldsymbol{x})$, a noisy estimator of $\nabla_{\boldsymbol{x}} \boldsymbol{f}(\boldsymbol{x})$ such that $\nabla_{\boldsymbol{x}} \hat{\boldsymbol{f}}(\boldsymbol{x}) = \nabla_{\boldsymbol{x}} \boldsymbol{f}(\boldsymbol{x}) + \boldsymbol{\varepsilon}$ with $\boldsymbol{\varepsilon} \in \mathbb{R}^{W \times H}$. We denote $\|\cdot\|_F$ as the Frobenius norm.*

So far, we do not consider this noise to be random, and we aim at bounding the leftover noise after the application of our method. In particular, we'll notice that at the level $\sigma^*$, it only depends on the value of $\sigma^*$ and is always upper bounded by the norm of the original noise.

**Proposition A.2.** *Let $\nabla \hat{\boldsymbol{f}} = \nabla \boldsymbol{f} + \boldsymbol{\varepsilon}$ as the noisy gradient of $\boldsymbol{f}$, with $\boldsymbol{\varepsilon} \in \mathbb{R}^{W \times H}$. For $\sigma^* = \inf \left\{ \sigma : \|\mathcal{F}(\nabla \boldsymbol{f}) \odot \bar{\boldsymbol{K}}^\sigma\|_F^2 = 0 \right\}$, we have*

$$\|\mathcal{F}^{-1}(\mathcal{F}(\nabla \hat{\boldsymbol{f}}) \odot \boldsymbol{K}^{\sigma^*}) - \nabla \boldsymbol{f}\|_F^2 = \|\mathcal{F}^{-1}(\mathcal{F}(\boldsymbol{\varepsilon}) \odot \boldsymbol{K}^{\sigma^*})\|_F^2 \leq \|\boldsymbol{\varepsilon}\|_F^2, \tag{1}$$

*where $\odot$ is the Hadamard product, $\boldsymbol{K}^{\sigma^*}$ a binary mask for low-pass filtering of frequency $\sigma$, and $\bar{\boldsymbol{K}}^{\sigma^*}$ is the opposite mask.*

Submitted to 37th Conference on Neural Information Processing Systems (NeurIPS 2023). Do not distribute.

*Proof.*

$$\|\mathcal{F}^{-1}(\mathcal{F}(\nabla \hat{\boldsymbol{f}} \odot \boldsymbol{K}^\sigma)) - \nabla \boldsymbol{f}\|_F^2 = \|\mathcal{F}^{-1}(\mathcal{F}(\nabla \hat{\boldsymbol{f}} \odot \boldsymbol{K}^\sigma)) - \mathcal{F}^{-1}(\mathcal{F}(\nabla \boldsymbol{f}))\|_F^2 \tag{2}$$

$$= \|\mathcal{F}^{-1}(\mathcal{F}(\nabla \boldsymbol{f} + \boldsymbol{\varepsilon} \odot \boldsymbol{K}^\sigma)) - \mathcal{F}^{-1}(\mathcal{F}(\nabla \boldsymbol{f}))\|_F^2 \tag{3}$$

$$= \|\mathcal{F}^{-1}(\mathcal{F}(\nabla \boldsymbol{f} \odot \boldsymbol{K}^\sigma) + \mathcal{F}(\boldsymbol{\varepsilon} \odot \boldsymbol{K}^\sigma)) - \mathcal{F}^{-1}(\mathcal{F}(\nabla \boldsymbol{f}))\|_F^2 \tag{4}$$

$$= \|\mathcal{F}^{-1}(\mathcal{F}(\nabla \boldsymbol{f} \odot \boldsymbol{K}^\sigma) + \mathcal{F}(\boldsymbol{\varepsilon} \odot \boldsymbol{K}^\sigma) - \mathcal{F}(\nabla \boldsymbol{f}))\|_F^2 \tag{5}$$

$$= \|\mathcal{F}^{-1}(\mathcal{F}(\boldsymbol{\varepsilon} \odot \boldsymbol{K}^\sigma) - \mathcal{F}(\nabla \boldsymbol{f} \odot \bar{\boldsymbol{K}}^\sigma))\|_F^2 \tag{6}$$

$$\tag{7}$$

17 Then by selecting $\sigma^* = \in \boldsymbol{f} \left\{ \sigma : \|\mathcal{F}(\nabla \boldsymbol{f} \odot \bar{K}^\sigma) - \mathcal{F}(\nabla \boldsymbol{f})\|_F^2 = 0 \right\}$, we have that

$$\|\mathcal{F}^{-1}(\mathcal{F}(\boldsymbol{\varepsilon} \odot \boldsymbol{K}^\sigma) - \mathcal{F}(\nabla \boldsymbol{f} \odot \bar{K}^\sigma))\|_F^2 = \|\mathcal{F}^{-1}(\mathcal{F}(\boldsymbol{\varepsilon}) \odot \boldsymbol{K}^\sigma)\|_F^2 \tag{8}$$

$$\leq \|\boldsymbol{\varepsilon}\|_F^2 \tag{9}$$

$$\tag{10}$$

18 $\qquad\square$

19 Now, by adding an assumption of randomness on the noise $\varepsilon$, we can deduct the distribution of the
20 ratio of the two last members of the previous proposition, and deduce an order of scale of the norm of
21 the remaining noise after filtration, compared to its original norm.

22 **Proposition A.3.** *Let the noise $\boldsymbol{\varepsilon} \in \mathbb{R}^{W \times H}$ follow a normal distribution $\boldsymbol{\varepsilon} \sim \mathcal{N}(0, \varsigma)^{\otimes WH}$. Then*
23 *the norm of the Fourier spectra of the noise $\|\mathcal{F}(\boldsymbol{\varepsilon})\|_F^2 \sim \Gamma(k = 2WH, \theta = \varsigma^2 WH)$ and filtered*
24 *noise $\|\mathcal{F}(\boldsymbol{\varepsilon}) \odot \boldsymbol{K}^\sigma\|_F^2 \sim \Gamma(k = 8\sigma^2, \theta = 4\varsigma^2\sigma^2)$ follows Gamma distributions.*

25 *Therefore, the ratio of the two distributions $R = \|\mathcal{F}(\boldsymbol{\varepsilon})\|_F^2 / \|\mathcal{F}(\boldsymbol{\varepsilon}) \odot \boldsymbol{K}^\sigma\|_F^2$ follows a Beta prime*
26 *distribution $R \sim \beta'\left(2WH, 4\sigma^2, 1, \frac{WH}{4\sigma^2}\right)$.*

27 *Proof.* As defined previously, $\boldsymbol{\varepsilon}_{(i,j)} \sim \mathcal{N}(0, \varsigma)$. It is well known that the Fourier transform of that
28 random matrix follows a complex normal distribution, or equivalently the real and imaginary random
29 variables of the Fourier transform are i.i.d normally distributed, as $\mathcal{RF}(\boldsymbol{\varepsilon})_{(i,j)} \sim \mathcal{N}(0, \varsigma\sqrt{WH/2})$
30 and $\mathcal{IF}(\boldsymbol{\varepsilon})_{(i,j)} \sim \mathcal{N}(0, \varsigma\sqrt{WH/2})$. Therefore the scaled norm follows a $\chi^2$ distribution.

$$\frac{\|\mathcal{F}(\boldsymbol{\varepsilon})\|_F^2}{\varsigma\sqrt{WH/2}} \sim \chi^2_{2WH} \qquad \text{and equivalently,} \qquad \frac{\|\mathcal{F}(\boldsymbol{\varepsilon}) \odot \boldsymbol{K}^\sigma\|_F^2}{\varsigma\sqrt{WH/2}} \sim \chi^2_{8\sigma^2}. \tag{11}$$

31 Note that $\chi^2$ distributions are a specific case of Gamma distributions, and can therefore be noted as
32 follows $\|\mathcal{F}(\boldsymbol{\varepsilon})\|_F^2 \sim \Gamma(k = WH, \theta = \varsigma^2 WH)$ and filtered noise $\|\mathcal{F}(\boldsymbol{\varepsilon}) \odot \boldsymbol{K}^\sigma\|_F^2 \sim \Gamma(k = 4\sigma^2, \theta =$
33 $4\varsigma^2\sigma^2)$. Finally, let $R = \|\mathcal{F}(\boldsymbol{\varepsilon})\|_F^2 / \|\mathcal{F}(\boldsymbol{\varepsilon}) \odot \boldsymbol{K}^\sigma\|_F^2$. Since both random variables follow a Gamma
34 distribution, their ratio is a Beta prime distribution of parameters $R \sim \beta'\left(WH, 4\sigma^2, 1, \frac{WH}{4\sigma^2}\right)$.

35 The results remain valid in the input space by Parseval's Theorem [1], up to a constant factor. $\qquad\square$

36 **Proposition A.4.** *We recall that SmoothGrad is defined as $SG = \frac{1}{n} \sum_{i=1}^n \nabla_{\boldsymbol{x}} \boldsymbol{f}(\boldsymbol{x} + \boldsymbol{\delta}_i)$ with*
37 *$\forall_{i=1,\dots,n} \boldsymbol{\delta}_i \in \mathcal{N}(0, \varsigma)^{\otimes WH}$. $\nabla_{\boldsymbol{x}} \boldsymbol{f}(\boldsymbol{x} + \boldsymbol{\delta}_i)$ is a random matrix and therefore $SG$ is. Assuming our*
38 *predictor $\boldsymbol{f} \in L\text{-}Lip(\mathcal{X})$ is L-Lipschitz. We denote $\|\cdot\|_2$ as the spectral norm, and define the variance*
39 *as $\mathbb{V}(SG) = \max\left(\|\mathbb{E}((SG - \mathbb{E}SG) \cdot (SG - \mathbb{E}SG)^T)\|_2, \|\mathbb{E}((SG - \mathbb{E}SG)^T \cdot (SG - \mathbb{E}SG))\|_2\right)$.*
40 *We then have, for $t > 0$,*

$$\mathbb{P}\left(\|SG - \mathbb{E}SG\|_2 \geq t\right) \leq (W + H) \cdot \exp\left(\frac{-t^2 n^2 / 2}{\mathbb{V}(SG) + 2Lt/3}\right). \tag{12}$$

41 *Proof.* We first demonstrate the bounded difference property

$$\forall_{i=1\cdots n} \|\nabla_{\boldsymbol{x}} \boldsymbol{f}(\boldsymbol{x} + \boldsymbol{\delta}_i) - \mathbb{E}\nabla_{\boldsymbol{x}} \boldsymbol{f}(\boldsymbol{x} + \boldsymbol{\delta}_i)\|_2 \leq \|\nabla_{\boldsymbol{x}} \boldsymbol{f}(\boldsymbol{x} + \boldsymbol{\delta}_i)\|_2 + \|\mathbb{E}\nabla_{\boldsymbol{x}} \boldsymbol{f}(\boldsymbol{x} + \boldsymbol{\delta}_i)\|_2 \leq 2L. \tag{13}$$

42 Finally, the result follows by application of the Matrix Bernstein Inequality [2]. $\qquad\square$

## B  Different measures of complexity

To corroborate the results in S3.1 and S3.3 computing the high-frequency content from Kolmogorov image compression metric, we reproduce those experiments using the Laplace2-operator. We additionally show the evolution of high-frequency content in VGG16 and ViT using both metrics. On all the plots, we observe a similar trend between the metrics, confirming that :

- Gradient-based and prediction-based methods have a different signature in the Fourier spectrum (see Figure 1)
- On both convolutional models (ResNet50 and VGG) we observe more high frequencies when the models contain an MaxPooling operation or strided convolution instead of Aver-agePooling (see Figures 2, 3, 3, top curves)
- On both convolutional models, training the model doesn't alleviate the high frequency content (see Figures 2, 3, 3, bottom curves). However, we observe a different behaviour on ViT, suggesting that a different mechanism is responsible for the high-frequency content in this kind of architectures (see Figures 5, 6).

### B.1  For the methods

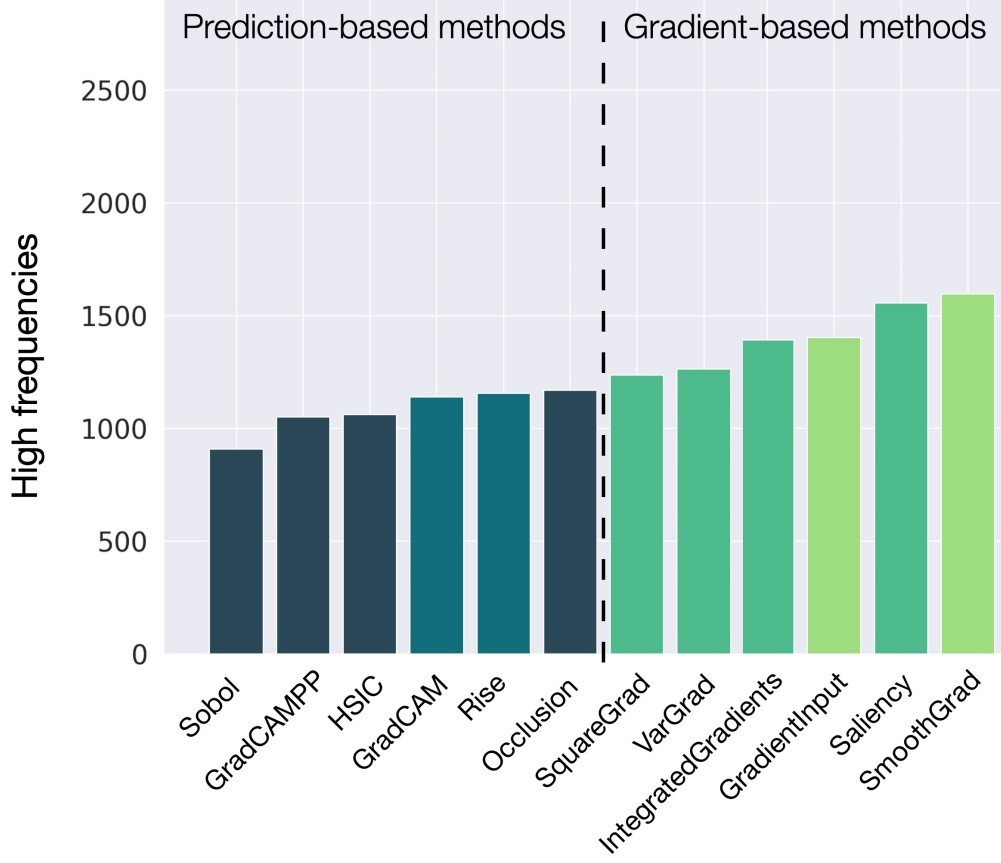

Figure 1: **High-frequency power in attribution methods.** High-frequency power present in the importance maps derived from different attribution methods, computed using Laplacian2 operator. Prediction-based methods produce less high-frequency content than gradient-based methods.

 **B.2 For the models**

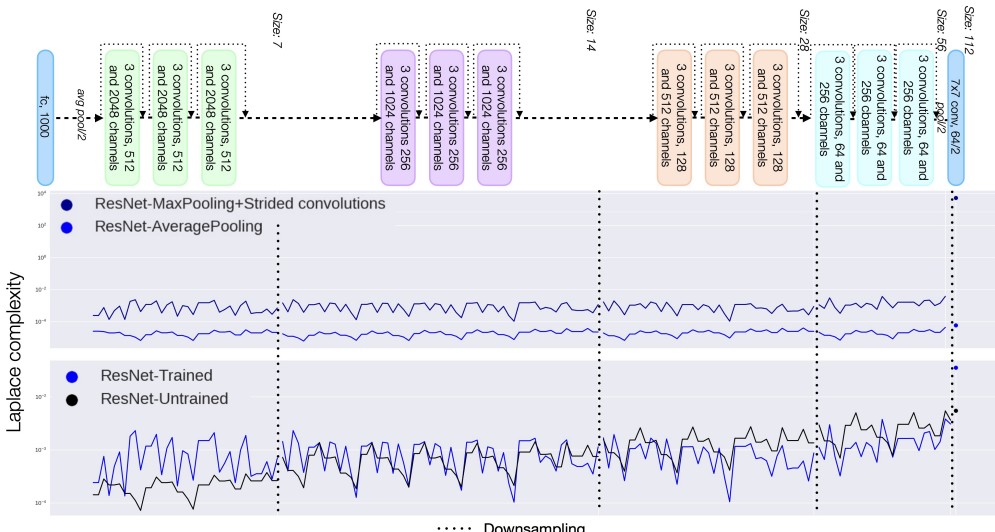

Figure 2: **Evolution of the high-frequency content in Resnet50.** We compute the high-frequency content along the depth of a ResNet50 varying either the weights or the pooling, using Laplace2-operator. The top curve illustrates the impact of different poolings, with MaxPooling and stride shown in dark blue and AveragePooling in light blue. The bottom curve represents the trained model, indicated by the blue curve, while the untrained model is represented by the black curve. Each point on the graph corresponds to a layer within the model.

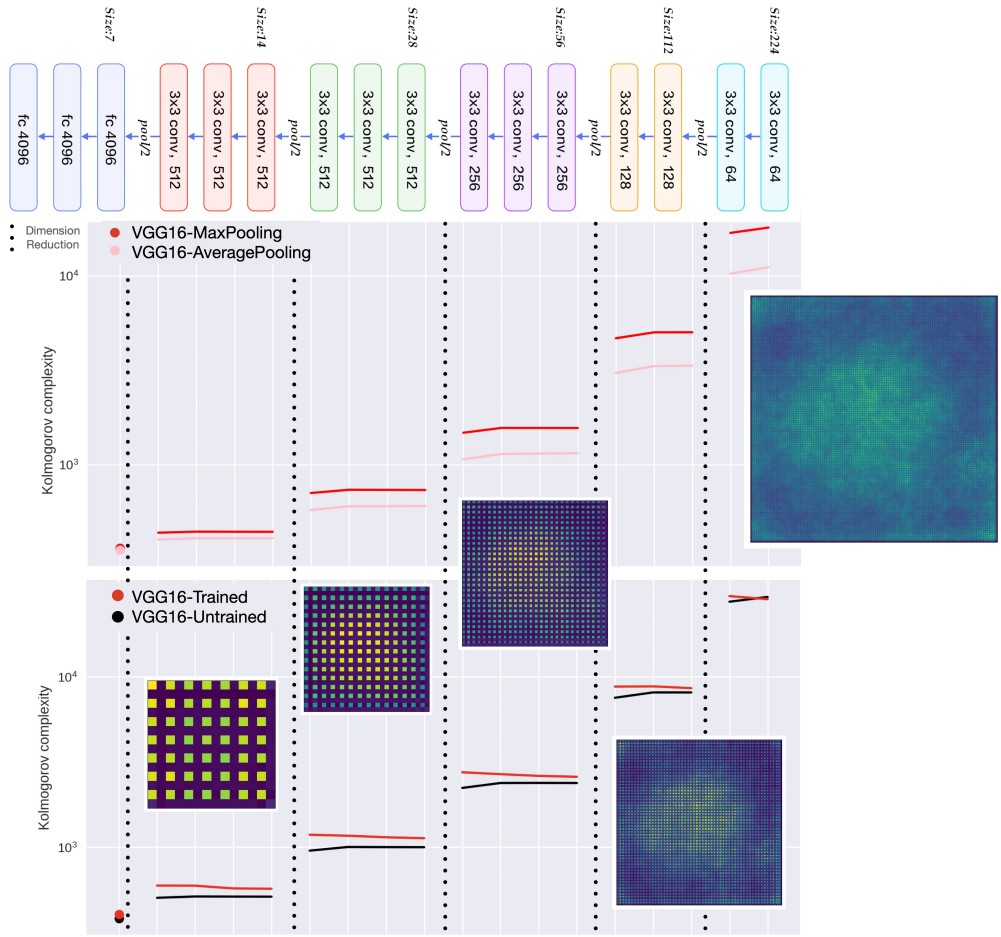

Figure 3: **Evolution of the high-frequency content in VGG16 with Kolmogorov complexity.** We compute the high-frequency content along the depth of a VGG16 varying either the weights or the pooling, using Kolmogorov image compression. The top curve illustrates the impact of different poolings, with MaxPooling shown in dark red and AveragePooling in pink. The bottom curve represents the trained model, indicated by the red curve, while the untrained model is represented by the black curve. Each point on the graph corresponds to a layer within the model.

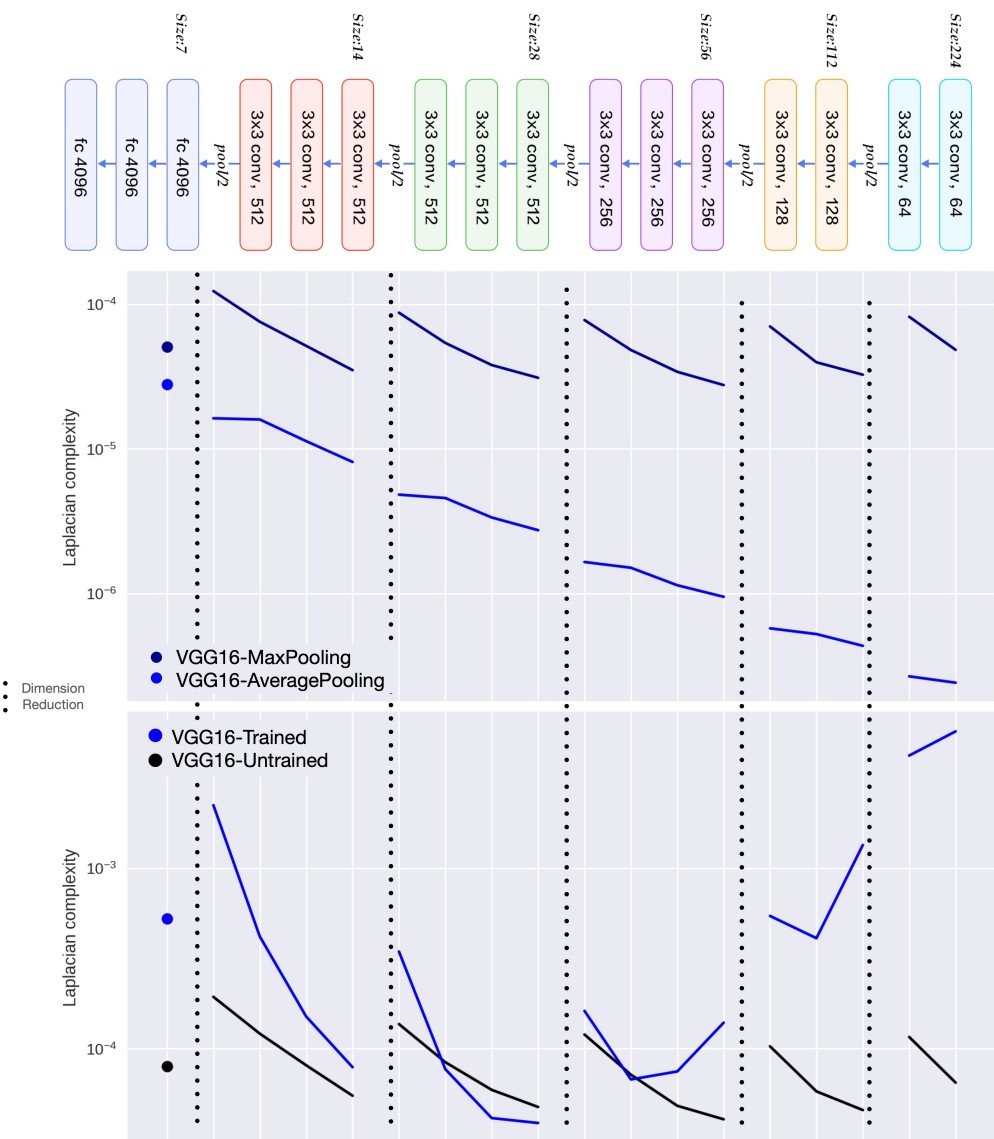

Figure 4: **Evolution of the high-frequency content in VGG16 with Laplace complexity.** We compute the high-frequency content along the depth of a VGG16 varying either the weights or the pooling, using Laplace2-operator. The top curve illustrates the impact of different poolings, with MaxPooling shown in dark blue and AveragePooling in light blue. The bottom curve represents the trained model, indicated by the blue curve, while the untrained model is represented by the black curve. Each point on the graph corresponds to a layer within the model.

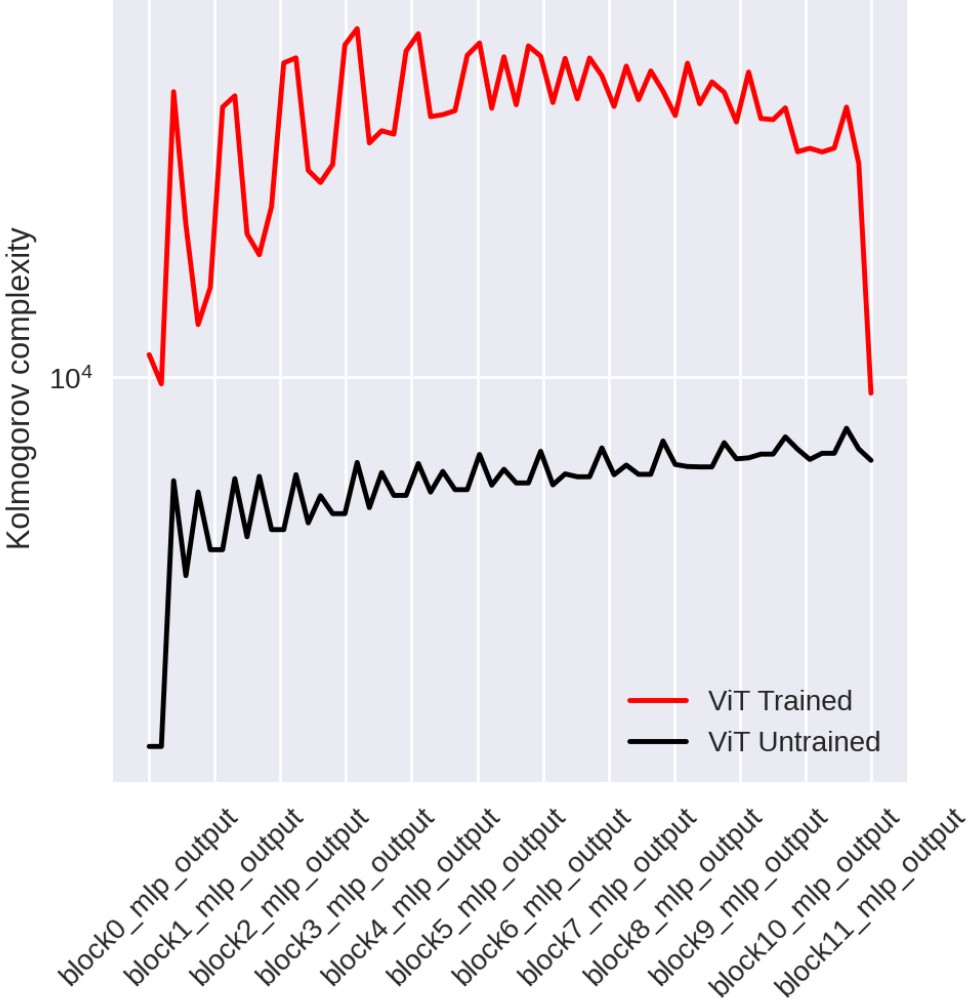

Figure 5: **Evolution of the high-frequency content in ViT with Kolmogorov complexity.** We compute the high-frequency content along the depth of a ViT varying the weights, using Kolmogorov image compression. The curve represents the trained model, indicated by the red curve, while the untrained model is represented by the black curve. Each point on the graph corresponds to a layer within the model. We show the labels representing the end of a block to give an idea of the evolution of the complexity inside a block.

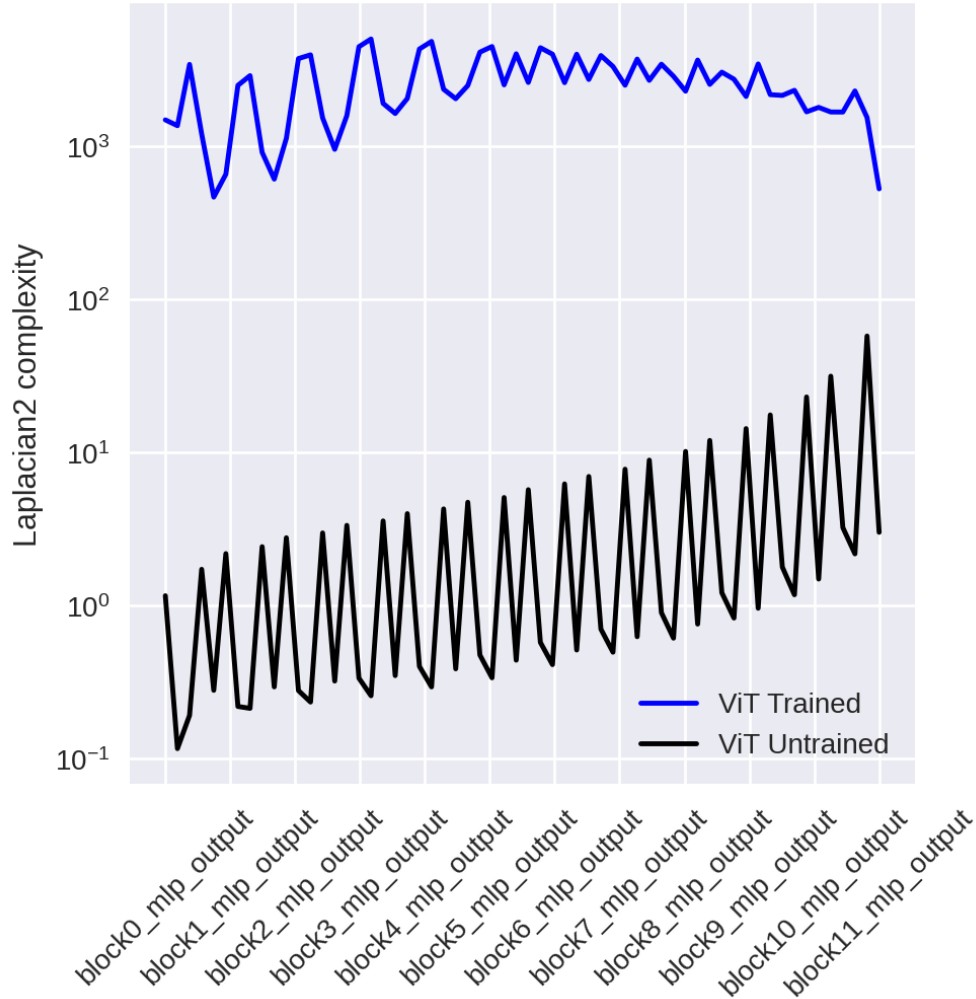

Figure 6: **Evolution of the high-frequency content in ViT with Laplace complexity.** We compute the high-frequency content along the depth of a ViT varying the weights, using Laplace2-operator. The curve represents the trained model, indicated by the blue curve, while the untrained model is represented by the black curve. Each point on the graph corresponds to a layer within the model. We show the labels representing the end of a block to give an idea of the evolution of the complexity inside a block.

## C  Control conditions for the evidence of noise in the gradient

We add several control conditions on the experiment S3.2 to show the difference between an informative and uninformative gradient. We propose three different approaches :

- $\sigma = 0 \equiv \nabla_{\boldsymbol{x}}^{\sigma} \boldsymbol{f}(\boldsymbol{x}) = 0$ . In that case, we measure $||\boldsymbol{f}(\boldsymbol{x} + \epsilon) - \boldsymbol{f}(\boldsymbol{x})||_2$

- $\nabla_{\boldsymbol{x}}^{C} \boldsymbol{f}(\boldsymbol{x}) = \rho(\nabla_{\boldsymbol{x}} \boldsymbol{f}(\boldsymbol{x}))$ where $\rho$ represents the permutation operator. Here we destroy the spatial structure, and therefore the information represented by high and low frequencies.

- $\nabla_{\boldsymbol{x}}^{C} \boldsymbol{f}(\boldsymbol{x}) \sim \mathcal{U}(min(\nabla_{\boldsymbol{x}} \boldsymbol{f}(\boldsymbol{x})), max(\nabla_{\boldsymbol{x}} \boldsymbol{f}(\boldsymbol{x})))$. Finally, we measure the information carried by some random noise, following a uniform distribution.

In the three cases, we compute the L2 norm between the first-order approximation of the model and something else containing no relevant information. We therefore expect a resulting curve with a higher estimation error than the others, containing some relevant information. We observe that is always the case for ResNet and ViT. This observation is a bit less clear for some radius value in the two first conditions on ConvNeXT but is present in the third condition.

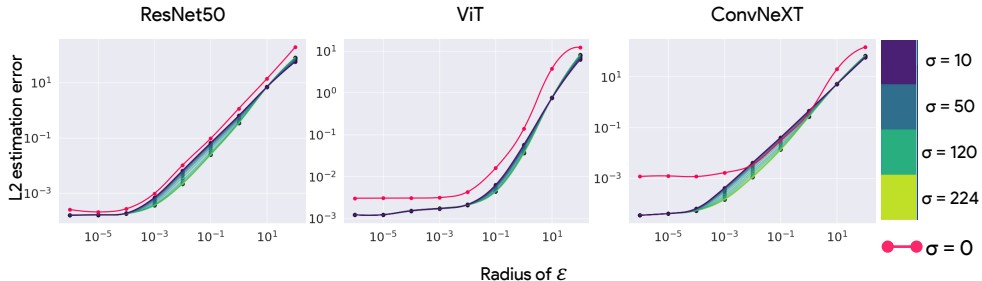

Figure 7: **Control condition $\sigma = 0$.** We plot the residual of the first-order approximation of the model, that is $\boldsymbol{f}(\boldsymbol{x} + \boldsymbol{\varepsilon}) \approx \boldsymbol{f}(\boldsymbol{x}) + \boldsymbol{\varepsilon} \nabla_{\boldsymbol{x}} \boldsymbol{f}(\boldsymbol{x})$, with the gradient $\nabla \boldsymbol{f}$ filtered at different bandwidths $\sigma$. Additionally, we plot the control condition $\sigma = 0$ in pink, representing the absence of information from the gradient.

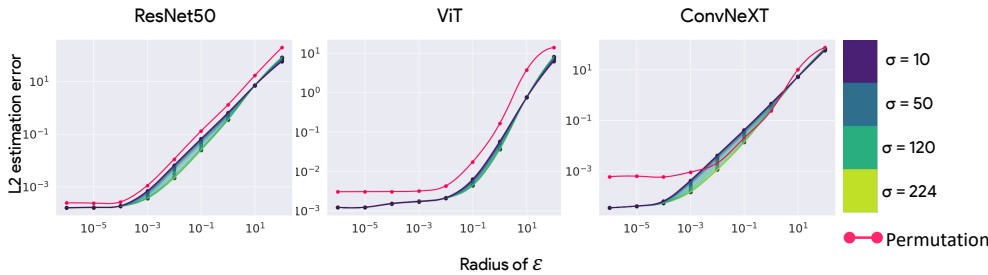

Figure 8: **Control condition $\nabla_{\boldsymbol{x}}^{C} \boldsymbol{f}(\boldsymbol{x}) = \rho(\nabla_{\boldsymbol{x}} \boldsymbol{f}(\boldsymbol{x}))$.** We plot the residual of the first-order approximation of the model, that is $\boldsymbol{f}(\boldsymbol{x} + \boldsymbol{\varepsilon}) \approx \boldsymbol{f}(\boldsymbol{x}) + \boldsymbol{\varepsilon} \nabla_{\boldsymbol{x}} \boldsymbol{f}(\boldsymbol{x})$, with the gradient $\nabla \boldsymbol{f}$ filtered at different bandwidths $\sigma$. Additionally, we plot the control condition $\nabla_{\boldsymbol{x}}^{C} \boldsymbol{f}(\boldsymbol{x}) = \rho(\nabla_{\boldsymbol{x}} \boldsymbol{f}(\boldsymbol{x}))$ in pink, representing some unstructured information from the gradient.

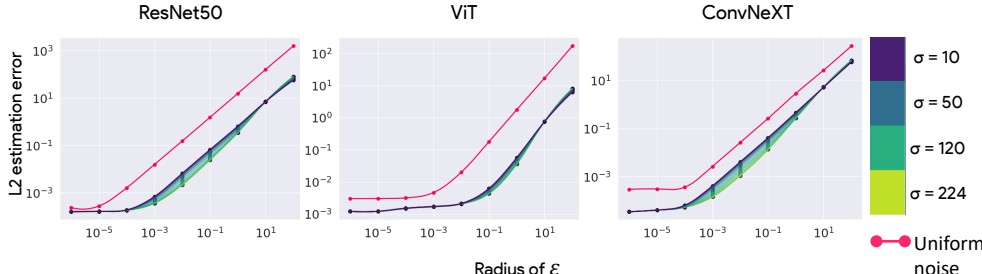

Figure 9: **Control condition** $\nabla_{\boldsymbol{x}}^{C} \boldsymbol{f}(\boldsymbol{x}) \sim \mathcal{U}(min(\nabla_{\boldsymbol{x}} \boldsymbol{f}(\boldsymbol{x})), max(\nabla_{\boldsymbol{x}} \boldsymbol{f}(\boldsymbol{x})))$. We plot the residual of the first-order approximation of the model, that is $\boldsymbol{f}(\boldsymbol{x} + \boldsymbol{\varepsilon}) \approx \boldsymbol{f}(\boldsymbol{x}) + \boldsymbol{\varepsilon}\nabla_{\boldsymbol{x}} \boldsymbol{f}(\boldsymbol{x})$, with the gradient $\nabla \boldsymbol{f}$ filtered at different bandwidths $\sigma$. Additionally, we plot the control condition $\nabla_{\boldsymbol{x}}^{C} \boldsymbol{f}(\boldsymbol{x}) \sim \mathcal{U}(min(\nabla_{\boldsymbol{x}} \boldsymbol{f}(\boldsymbol{x})), max(\nabla_{\boldsymbol{x}} \boldsymbol{f}(\boldsymbol{x})))$ in pink, representing some random information from the gradient.