# OpenReview forum: "Gradient strikes back: How filtering out high frequencies improves explanations"
_NeurIPS.cc/2023/Conference — Submitted to NeurIPS 2023_

### Official Review · Reviewer_QucS · 2023-06-22

**Soundness:** 3 good
**Presentation:** 3 good
**Contribution:** 2 fair
**Rating:** 5
**Confidence:** 4

**Summary:**

This paper investigates the issue of noisy gradient-based explanations caused by high-frequency content. The authors empirically identify that several down-sampling operations, such as MaxPooling or stride convolution, could be the primary source of these high frequencies. To address this, they present FORGrad, which removes high-frequency content in gradients by leveraging low-pass filter on the Fourier spectrum. The results on three faithfulness metrics further show the effectiveness of FORGrad in improving existing gradient-based explanation methods.

**Strengths:**

1. The paper is well-written, and the motivation is sound.
2. The analysis regarding the high-frequency noises in gradients is interesting and could benefit the further designs of explanation methods.
3. The proposed FORGrad achieves promising results on three faithfulness metrics, potentially improving existing gradient-based methods.


**Weaknesses:**

1. Throughout the paper, the term “High frequencies” is extensively employed to describe the phenomenon in gradients. However, how to clearly define “high” in actual deployment is not well explained. This could become especially problematic considering the significant variations in model architectures and datasets, thus degrading the significance of this work.
2. The argument, “High frequencies are just noise in the gradient,” is too strong. The analysis in Section 3.2 can only show that the high-frequency information has less effect on model decisions. This does not necessarily mean that such high-frequency information is always noise. Moreover, the perturbation-based analysis also has many limitations, e.g., the OOD problem [1]. It’d be better if the authors could provide another set of experiments to validate their findings.
3. It is nice to see that FORGrad could co-work with many gradient-based methods. However, in many cases, the improvement introduced by FORGrad is minor/marginal, e.g., Int.Grad* on ConvNeXT. It thus remains unclear whether FORGrad is genuinely effective or merely a result of randomness.
4. The source of high-frequency gradients is not systematically investigated. The analysis in Section 3.3 only covers two mechanisms, i.e., downsampling and training. This narrow focus fails to provide substantial evidence for the claim that these mechanisms are the main reasons, as other relevant factors are not adequately discussed.

[1] Peter Hase, Harry Xie, Mohit Bansal. The Out-of-Distribution Problem in Explainability and Search Methods for Feature Importance Explanations. NeurIPS 2021.


**Questions:**

1. Grad-CAM and Grad-CAM++ should be categorized as gradient-based methods?
2. Does the parameter $\sigma$ have a consistent value across all explanation methods, model architectures, and layers? If not, how can we determine the optimal value?

---

> ### Author Rebuttal · Authors · 2023-08-09
>
> The reviewer's dedicated time and their kind appraisal of our work are greatly appreciated by us. Each of the weaknesses noted by the reviewer has been addressed in the following content, as detailed in our responses.
>
> **- 1:** Thank you for this insightful comment and for carefully checking the details in this work.
> We agree that the notion of “high-frequency” might have not been sufficiently spelled out. We have edited the manuscript to clarify this definition.
>
> To summarize, FORGrad consists in selecting an optimal cut-off for low-pass filtering the gradient. We call the filtered-out content high-frequency. According to our observations, the optimal sigma value is consistent across models. Therefore, the notion of high frequency could be associated with the average amount of information filtered by FORGrad.
> We also added a figure as supplementary material to show the histogram of frequencies present in the explanations with the average sigma value across models to visualize the ratio of high/low frequencies.
>
> **- 2:** In the specified section, the language has been adjusted to be less assertive. The objective of this section is to offer empirical validation for the presence of noise by illustrating that we can reasonably exclude high frequencies due to their negligible information content. We would also like to point the reviewer to the appendix for additional results from three experiments, which highlight the effect of only low-frequency content versus purely noisy information. These results can serve as additional experiments highlighting the sufficiency of low-frequency content on model decisions and justify the decision to consider high-frequency content as non-necessary, therefore removable.
>
> **- 3:** We are grateful for the positive feedback regarding our method, and we acknowledge that in certain instances, the improvement may be perceived as marginal. Nonetheless, FORGRad proves beneficial in the vast majority of cases (even leading to a complete reversal of rankings in the case of resnet50!) while incurring such a negligible cost that we argue it justifies the broad adoption of this approach. When combined with techniques like Saliency or SmoothGrad, our method offers a highly efficient and straightforward means to improve explanations from models – effectively competing with the latest state-of-the-art methodologies.
> Additionally, as the improvement is consistent across all methods and all three models, we argue that this cannot be due to a random effect but rather an actual improvement in the quality of the explanation.
>
> In fact, for each metric we performed a Bayesian ANOVA (see details below). Confirming our hypothesis, we found a strong difference in the BEFORE-AFTER factor (comparing scores before and after FORGrad) in the INS metric (BEFORE-AFTER factor BF10=11.33, error= 1%), and in the FID metric (BEFORE-AFTER factor BF10=118.65, error= 2%), and moderate evidence for the DEL metric (BEFORE-AFTER factor BF10=1.79, error= 2%). All in all, considering the small number of samples used for this analysis, we are very confident that the results of the statistical analysis confirm a significant difference between the scores obtained before and after applying FORGrad on the different attribution methods.
>
> Details on Bayesian ANOVA analysis: We analyzed the metrics by means of Bayesian ANOVAs, considering BEFORE-AFTER and ATTRIBUTION-METHODS as fixed factors, and MODELS as a random term. In all analyses, we computed Bayes Factors (BF) as the ratio between the models testing the alternative against the null hypothesis. The alternative hypothesis states that the factor is important for explaining the variance in the data (i.e., there is a significant difference between conditions, for example before and after). All BFs are denoted as BF10. In practice, BFs provide substantial (BF>\~5) or strong (BF>\~10) evidence in favor of the alternative hypothesis, and low BF (BF<~0.5) suggests a lack of effect (Masson, 2011).
>
> To further validate FORGrad, we augmented the table of results with the standard deviation of the scores for all the metrics and we added the results of our statistical test in the main text.
>
> **- 4:** We agree that we have not investigated in detail all of the possible mechanisms possibly contributing to the presence of noise in the gradients. As mentioned in the general comment, we chose to focus on the ones that seem to be mainly responsible for the noise according to our observations and provide evidence for it in the paper. This is however not the main contribution of the work.
>
> The additional experiment with the different striding also supports our hypothesis and provides a first step toward a more complete understanding of each specific mechanism in the models and their effect on the gradients.
>
> *Grad-CAM and Grad-CAM++ should be categorized as gradient-based methods?* You are absolutely right, we now have corrected this in the figures and the document.
>
> *Does the parameter sigma have a consistent value across all explanation methods, model architectures, and layers? If not, how can we determine the optimal value?* We currently compute the optimal sigma per model and keep its value constant across all explanation methods. We propose FORGrad as a very flexible method where the optimal sigma can be common across all models/datasets/methods but can also be specific with respect to all those criteria. As detailed in section 3.4, the sigma is computed by selecting the optimal cut-off frequency. Its optimal value corresponds to the one with the best score according to those metrics.
>
> However, you are correct, and a possible extension of this work would be to adapt the sigma per model/method/dataset. We added this possibility to the discussion.

---

> > ### Comment · Reviewer_QucS · 2023-08-15
> >
> > I want to thank the authors for the detailed response. Most of my concerns have been well addressed, especially Weakness 3. But as discussed, the paper still has limitations on the choice of sigma and the in-depth analysis of high-frequency gradients.
> >
> > Overall, I can see that the strengths outweigh the weaknesses of the current version. I decided to raise my score to "borderline accept." Related to this, the authors must include detailed numbers (Weakness 3) and revise the explanations (Weakness 1&2) in the main paper.

---

> > > ### Author Response · Authors · 2023-08-21
> > >
> > > We thank the reviewer for this kind comment. As asked, we included the detailed numbers and revised explanations in the paper.

---

### Official Review · Reviewer_bkGG · 2023-06-29

**Soundness:** 2 fair
**Presentation:** 3 good
**Contribution:** 2 fair
**Rating:** 5
**Confidence:** 4

**Summary:**

The paper analyzes the two types of attribution methods: gradient-based/ white-box methods and prediction-based/ black-box methods. The authors observe that the faithfulness of black-box methods surpasses the white-box methods despite white-box methods accessing the internals of the classifier being explained. The authors suspect the presence of noise in the gradients to be a contributing factor to its limited faithfulness. They further perform Fourier spectral analysis to examine the distribution of attribution signal frequencies. This analysis sheds light on the existence of high-frequency noise signals in the gradients, a by-product of pooling and stride operations, which are propagated as attribution signals. The obtained Fourier spectrum is filtered using a low-pass filter with an optimal cut-off threshold learned based on optimizing the Insertion and Deletion metrics, which are proxies for the faithfulness of the explanations. This yields a faithful explanation from the white-box methods surpassing the faithfulness that black-box methods achieved earlier.

**Strengths:**

* The methodology is effectively presented, incorporating suitable figures and accompanying text that elucidate the significance of these visuals. The paper follows a systematic approach in introducing concepts, starting with the motivation, followed by the hypothesis, supporting evidence, and ultimately presenting a solution to the problem. This coherent structure enhances the elegance of the methodical exposition.

* The analysis of the gradient signals is sound, and the link between noise in the gradient signals and explainability is adequately established.

* The solution’s strengths are its simplicity - performing a FFT and identifying a suitable frequency threshold; and modularity - can be integrated with many whitebox explainability approaches. It is a simple post-processing strategy to extract high fidelity explanations.

* The experiments are somewhat comprehensive - with comparison against some of the prevalent explainability approaches.


**Weaknesses:**

- *Possible misrepresentation of the related literature:* Reference [3] is described as both a black-box method and a white-box method in different sections. This inconsistency is also reiterated in Line 76. Similarly, in Table 1, Grad CAM and Grad CAM++ are categorized as prediction-based/black-box methods despite their reliance on gradients. These methods employ gradient backpropagation (typically) up to the last convolution layer to generate the CAM, which contradicts the authors’ characterization as purely black-box approaches.

- *Missing related literature and experiments:* The analysis of attribution-based explainability approaches overlooks significant contributions that introduce litmus tests for these methods, such as the work by Adebayo et al. NeurIPS 2018 and Sixt et al. ICML 2020. Sixt et al. explores the convergence of activations in saliency maps of gradient-based techniques when subjected to litmus tests proposed by Adebayo et al. It would be intriguing to investigate potential parallels between the convergence analysis presented by Sixt et al. and the spectral analysis conducted in the current paper. The application of Adebayo et al.'s litmus tests are necessary, and a comparative evaluation of the results before and after applying the proposed frequency cut-off indicating an increase in the faithfulness of the explanations could further strengthen the contribution. This is important considering the sweeping claim of the paper that gradient based attribution is reliable.

- *Missing experiments:* While the authors’ have conducted quantitative evaluation to highlight the effectiveness of the solution, the absence of qualitative evaluation is stark. It is evident from the Figures that the attribution map generated looks ‘better’ after the frequency cutoff. However, whether this attribution map measures up against other approaches (both white box and black box approaches) qualitatively needs to be checked. Human subject experiments to this effect are necessary.

- *Disconnected theoretical analysis:* The theoretical analysis seems disconnected from the empirical decisions in the current draft. The need for the theoretical analysis has to be well-motivated through a thorough rewrite.

**Minor suggestions**
- References [1] and [2] are just the same, with just a difference in the year. The author may need to check their bib source to remove the irrelevant bib entry and modify the paper just to retain the relevant entry.
- Line 84 activations have been mistyped as activities.
- Line 96 appears like a colloquial lingo. Please rewrite to make the statement formal to the research community.
- Please check for possible typos in Line 244.


**Questions:**

please see the weaknesses section

**Limitations:**

An analysis of the limitations has been honestly presented

---

> ### Author Rebuttal · Authors · 2023-08-09
>
> We extend our appreciation to the reviewer for sparing their time and for the thoughtful remarks they've provided regarding our work. We have taken the opportunity to respond to the weaknesses raised by the reviewer in the content that follows.
>
> **- Possible misrepresentation of the related literature:** Thank you for pointing out this mistake, we confirm that Reference [3] belongs to the black-box category and that GradCAM and GradCAM++ are white-box methods. We corrected this incoherence in the paper.
>
> **- Missing related literature and experiments:** The suggested references are indeed very relevant and we now cite those two papers in our article. Thank you for pointing out this missing literature.
> You are correct; we have incorporated qualitative and quantitative examples of Adebayo's sanity check in the rebuttal and confirm that ForGrad does not alter this characteristic. The methods that pass the sanity check continue to do so, while those that fail still do not meet the criteria.
>
> **- Missing experiments:** We agree that human experiments are valuable sanity checks in a field like explainability. However, it is not a standard procedure in the field. We, therefore, chose to use widely used and trusted metrics such as Deletion, Insertion, and Mu-Fidelity as done in representative work (Samek et al., 2021, Rong et al., 2022, Novello et al. 2022, Fel et al. 2021).
>
> **- Disconnected theoretical analysis:** We agree. As mentioned in the general comments, we have rewritten this section. We now better describe the connection between our two first propositions (on the norm of the gradient after low pass filtering, and on the distribution of the ratio of the norm of the error and the gradient) with the rest of the article. We have also clarified the fact that our third result (concerning the non-asymptotic convergence rate of SmoothGrad) should be regarded as additional and complementary to FORGrad. We only aim at demonstrating that SmoothGrad reduces the noise of the gradient of a given sample via several iterations, while FORGrad reduces this same noise in a single step.
>
> Finally, thank you for the suggestions on the text, we will correct those minor mistakes.

---

> > ### Comment · Reviewer_bkGG · 2023-08-16
> > **feedback to the authors**
> >
> > I thank the authors for providing responses to my queries and concerns. While this has improved my confidence in the paper, I am still not convinced with the significance of the observations. Hence, I am inclined to retain my initial rating.

---

> > > ### Author Response · Authors · 2023-08-21
> > >
> > > We thank the reviewer for the feedback and extend an invitation to read our response to reviewer vA9f, as we believe it provides some information that can justify the significance of our observations.

---

### Official Review · Reviewer_vA9f · 2023-07-01

**Soundness:** 2 fair
**Presentation:** 2 fair
**Contribution:** 2 fair
**Rating:** 3
**Confidence:** 4

**Summary:**

The paper explores the differences between prediction-based and gradient-based attribution methods for explaining the decisions of deep neural networks. The authors observe that these two approaches produce attribution maps with distinct power spectra, with gradient-based methods exhibiting more high-frequency content. This discrepancy raises questions about the origin and relevance of high-frequency information and why its absence in prediction-based methods leads to better explainability scores. By analyzing the gradient of visual classification models, the authors identify downsampling operations in Convolutional Neural Networks (CNNs) as a significant source of high-frequency content, potentially due to aliasing. To address this issue, they propose applying an optimal low-pass filter to improve gradient-based attribution methods, leading to enhanced explainability scores.


**Strengths:**

1. `Simple yet Effective Solution`: One of the key strengths of the paper is its proposal of a straightforward yet highly effective solution. By applying a frequency cut low-pass filter to the attribution maps, the authors successfully remove the high-frequency noise and improve the interpretability of the results. This simple intervention offers a practical "plug-and-play" approach that can provide meaningful and understandable attributions for human interpretation. The demonstrated enhancements in explainability scores serve as compelling evidence of the efficacy of this approach, showcasing its practical value in real-world applications.

**Weaknesses:**

1. `Flaw in Observation`: The paper's strong observation regarding the higher frequency content in gradient-based methods compared to prediction-based methods may be misleading. This difference is primarily attributed to the **distinct instances that each method focuses on, rather than being inherent properties of the attribution categories themselves**. Prediction-based methods typically concentrate on `image patches, super-pixels, or high-level features of specific receptive fields`, whereas gradient-based methods calculate derivatives with respect to each pixel individually. If gradients were calculated with respect to patches, super-pixels, or features, the resulting attributions would also exhibit dominance in low-frequency components.

2. `Naiveness in Theoretical Justification`: The paper's theoretical justification appears simplistic and lacks depth. It adopts a traditional signal processing approach, assuming the presence of noise with a specific form (such as uniform or Gaussian) and demonstrates the efficacy of a linear low-pass band cut filter. However, several critical questions remain unanswered. Firstly, the paper fails to address **why gradients with respect to the input are inherently noisy and exhibit high variance**. This noise can be attributed to the non-linearity and no-smoothness[A] of the network itself, and the downsampling operation might be just one contributing factor among others. Secondly, the derivation of the proposed solution seems to be a mere reproduction of standard textbook content on signal denoising, without providing any specific insights or considerations tailored to the problem at hand.

3. `Methodological Simplicity`: The application of a low-pass filter to gradients may be considered too simplistic for publication in a prestigious conference like NeurIPS. Similar techniques, such as those presented in reference [5] and other variance reduction methods, have already been established in the field. The paper does not introduce any novel or sophisticated methodologies, which might limit its overall contribution and impact.

[A] Wang Z, Wang H, Ramkumar S, et al. Smoothed geometry for robust attribution[J]. Advances in neural information processing systems, 2020, 33: 13623-13634.


**Questions:**

If you can assume a Gaussian for the gradient noise and has closed-form solution if noise scale ($\sigma$) is known, why still need to do hyper-parameter tuning in Figure 6? A more elegant solution should be estimate the scale and do it analytically.

---

> ### Author Rebuttal · Authors · 2023-08-09
>
> We express our thanks to the reviewer for allocating their time and for the encouraging feedback they've shared about our work.
> The ensuing content contains our responses to the weaknesses indicated by the reviewer.
>
> **- Flaw in Observation:** You are absolutely right, and it is indeed common knowledge that these methods reparameterize the explanation in a subspace, which leads to lower frequency content. However, it is important to emphasize that regardless, the conclusion remains unchanged: there are methods that explain with low frequencies and others with high frequencies, and this difference is accounted for by the class of methods used. It is precisely this fundamental distinction that drives us to introduce FORGrad and this provides a justification of why our method particularly improves gradient-based methods.
>
> **- Naiveness in Theoretical Justification:** We thank the reviewer for his detailed comment. First, a significant part of our article tackles this very question, as we have studied multiple architectures, and individual operations, in order to infer the inherent source of noise in the gradient. Second, while an analytical solution could be derived in principle, the problem turns out to be easier to solve in practice compared to our theoretical initial assumption, because we can discriminate the noise from the signal. In fact, we show experimentally that the form of the noise doesn’t matter and only resides in a specific part of the gradient (the high-frequency content). We however wanted to keep a general assumption in our theoretical section to make it applicable to any case.
>
> Additionally, our greedy algorithm, despite being simple, is more efficient at exploiting the real distribution of the noise as observed from the data.
>
> **- Methodological Simplicity:** As the old adage goes, never judge a book by its cover! FORGrad is simple – yet powerful and it addresses a crucial issue: gradient-based methods were efficient and powerful enough from the beginning; they are just susceptible to noise artifacts, which, as we show, can be easily alleviated. FORGrad yields SOTA results when applied to most gradient-based attribution techniques. Hence, we believe that it is likely to be widely used by the community.
>
> Unlike prediction-based methods such as RISE which require 8000 forward passes, FORGrad + gradient methods only require a single backward pass (!) – opening up real-world applications which could prove to be transformative for the industry. We also want to clarify that our work extends beyond merely low-pass filtering gradients. While this is a vital component of FORGrad, we also provide valuable insights into potential sources of noise and therefore open the door to new work aiming at alleviating this problem.
>
> *If you can assume a Gaussian for the gradient noise and has closed-form solution if noise scale (
> ) is known, why still need to do hyper-parameter tuning in Figure 6? A more elegant solution should be estimate the scale and do it analytically.*: Thank you for this suggestion. Indeed, knowing the standard deviation of the Gaussian could potentially lead to an analytical solution to solve the minimization of the L2 error between the noisy and filtered gradients. However, it is important to note that this approach does not provide us with any guarantees regarding the metric scores of faithfulness (Deletion, Insertion, Mufidelity).
>
> Moreover, we believe that our choice of assumption is conservative. Indeed, as our analysis suggests, the noise dominates the signal in high frequencies and it is minor or negligible elsewhere. However, our hypothesis assumes a uniformly weighted noise, frequency-wise. Therefore, while the approach of estimating the scale would be elegant, it would be too conservative, and our greedy algorithm allows us to leverage the noise information directly from our data.

---

> > ### Comment · Reviewer_vA9f · 2023-08-13
> > **Feedback to the author**
> >
> > Thank you for providing the updates. However, I still have ongoing concerns regarding both the **Theoretical Justification** and the proposed **Solution**.
> >
> > 1. `Theoretical Justification`: My main inquiry pertains to the fact that the issue highlighted in the paper appears to be more of an observation rather than a substantiated principle. In Section 3.2, the author observes that various architectures exhibit higher-order noise. Moving on to Section 3.3, the author identifies a specific operation (Max pooling) as the source of noise but does not establish the reasons behind this observation nor does the author delve into the noise generated by other operations. Consequently, the assertion of "universal high-frequency noise" in the paper appears to be more of a conjecture rather than a rigorously substantiated and mathematically supported conclusion. Moreover, the paper's proof merely demonstrates the advantageous nature of removing high frequencies to mitigate noise, which is very basic. Considering these factors, I am uncertain if the paper effectively addresses the issue it claims to tackle.
> >
> > 2. `Analytical solution VS hyper-parameter tuning`: The response provided by the author seems to deviate from my question. It might be helpful to include some baseline experiments in your work. Synthetic experiments, involving the introduction of known noise to the gradient, could also be conducted to evaluate whether the proposed proof truly tackles the problem. This is especially important as your assumption implies that the analytical solution should be optimally or at least nearly optimal. If the practical results do not align with this assumption, the entire derivation might be rendered invalid or at least diverge from the ultimate objective. I encourage you to validate this choice thoroughly.

---

> > > ### Author Response · Authors · 2023-08-21
> > >
> > > Thank you for acknowledging our rebuttal and providing an answer. However, we would like to provide some clarifications relative to your concerns.
> > >
> > >
> > > 1. `Theoretical Justification:`
> > >
> > > As stated in our rebuttal, our main contribution is to improve explainability methods. We provide computational evidence that high-frequency content reflects noise and not signal, coming from MaxPooling, as supported by the literature: (Zou et al, 2020), (Olah et al, 2017), (Vasconcelos et al. 2021) and  (Zhang, 2019), and justified by some visual observations: Figure 5  and explained by “the gradients (even averaged) following MaxPooling or strides exhibit checkerboard patterns, providing a plausible explanation for our quantitative observation of increased high-frequency content” (L170).
> > > Similarly, again in the general comment of our rebuttal, we provide some additional experiments showing the effect of other operations: skip connections and different values of stride. As mentioned in this comment, we don’t analyze in detail the precise contribution of each operation because the strided operations (such as MaxPooling) show a much more important effect than the others.
> > >
> > > Finally, it appears that we haven't used the phrase "universal high-frequency noise" within our paper. Our main assertion revolves around the idea that "high-frequencies contain noise." and proposes a simple method that achieves a state-of-the-art fidelity score for explainability.
> > >
> > > However, to strengthen those observations with some theoretical grounding, we added a demonstration in the supplementary material computing analytically the values of the lowest frequency in the worst-case scenario of the MaxPooling.
> > >
> > > 2. `Analytical solution VS hyper-parameter tuning:`
> > >
> > > Once again, our purpose is to improve the explanations, and our method guarantees to always provide an improvement, which is shown by the proof but also by our empirical results. As specified L236: “This result holds as long as we find σ∗”, we confirm that the result of our proofs depends on finding the optimal sigma – which we do empirically. Additionally, in L241 (“In that way, we demonstrate the always-positive effect of FORGrad on gradient methods”), we mention that FORGrad aims at improving the explanations, not finding the optimal solution.
> > >
> > > Introducing some known noise in the gradient would make the analytical solution easier to find – as we could have more information on the distribution of the noise and so adapt our assumption. But, it would have no implications on explainability, which is the focus of this work, the effective noise in the gradients of the explored models will very unlikely be of the same form as the one used for these tests. FORGrad would therefore remain a valid approach to improve gradient-based attribution methods.
> > >
> > > However, we are currently running some additional experiments with a toy model where we vary the operations and some parameters (such as the striding) and manipulate the gradient's content (as you suggested) to strengthen the link between the theoretical and experimental findings.

---

### Official Review · Reviewer_mo1q · 2023-07-07

**Soundness:** 3 good
**Presentation:** 3 good
**Contribution:** 3 good
**Rating:** 4
**Confidence:** 3

**Summary:**

The paper investigates why gradient-based attributions contain more high-frequencies compared to prediction-based
attribution. The authors argue that these high-frequency components are not relevant for the model and can
therefore seen as noise. Different saliency methods are analyzed on three different image models (ConvNeXT, ResNET, ViT). They further investigate the source of the noise and find that the downsampling-block is the the main cause for this.  Finally, the authors find that removing the noise from saliency maps yields an improved performance on
various explanation metrics.


**Strengths:**

- In general, I think the paper is novel. The noise in attribution maps and its effect on interpretability metrics
are not completly understood yet.
- The paper analyzes the causes of the noise in detail
- FORGrad makes it easy to interpolate between detailed and smooth attribution maps.
- The focus of the paper is clear and generally well executed.

**Weaknesses:**

- **Missing relevant work**: The work does not discuss the findings of [(Balduzzi et al. 2017)](http://proceedings.mlr.press/v70/balduzzi17b/balduzzi17b.pdf). This work investigates the effect of Residual connections on the noise of neural networks. Although this work has a focus more on saliency maps, the findings
of (Balduzzi et al. 2017) should be considered in the discussion.


- **How relevant are high-frequencies?** : "As anticipated, our observations reveal that the curves exhibiting reduced high-frequency content (from σ < 224 to σ = 10) closely align with the one of the non-filtered gradient (σ = 224)." (line 148). However, the distance between the curves σ < 224 and σ = 10 are quite substantial in Figure 4.
For ConvNeXT, the distance is almost one magnitude (the log-scale makes it look closer together).
Could you please report the maximum error between the curves in the supplementary material?

- **The conclusion is a bit off**:

  > (333) Overall, our work leads to a surprising result – that the almost forgotten gradient-based methods turn out to contain all the information needed to provide a faithful explanation of a model’s decision and that they can be as interpretable as the newest methods.

  I disagree with multiple points in the conclusion:
  - Gradient-based attribution almost forgotten? The integrated gradients paper received over 1000 citations last year (500 citations this year already). (source: semantic scholar)
  - I would be cautious about the faithfulness claim. It could also be that the metrics are biased and prefer a smoother attribution map.

- **Why not change the model?**: I would consider the raw gradient to be just the most faithful local representation of the model. If you prefer a smoother version, would it not make more sense to make changes to the underlying network architecture? This proposed methods feels like treating the symptoms instead of resolving the core issues.


To summarize, I am cautious about the main-take away that apply smoothing to gradient-based attribution methods makes them faithful. It would be great to also analyze if the faithfulness metrics are biased toward smooth attribuiton maps. Furthermore, I want to see the maximum error is between the different sigma-curves. It might be more significant
than suggested by the authors. A large error would questions the assumption that a smoothed gradient is more faithful.


**Questions:**

- How would you describe the relationship of your work to Balduzzi et al. 2017?

**Limitations:**


The limitation section reads very technical and misses a very important point: the Deletion, Insertion, and Fidelity metrics are only proxy metrics for interpretability. Performing well on them does not mean that the attribution map is actually interpretable. Only a user study could answer this question.

---

> ### Author Rebuttal · Authors · 2023-08-09
>
> Our sincere gratitude goes to the reviewer for their valuable time invested and for their generous appreciation of our efforts in this endeavor. In the subsequent content, we address each of the weaknesses highlighted by the reviewer.
>
> **- Missing relevant work:** Thank you for bringing this paper to our attention. It is now cited in the discussion (as you suggested). We answered the second part of the comment in the general section.
>
> **- How relevant are high-frequencies?** Thank you for this insightful comment and for reading carefully this section.
>
> The supplementary material, section C already provides an additional set of control experiments, showing different ways to compare our main results with a maximum error. We propose three ways to quantify the effect of the absence of information in the gradient on our measure. We reproduce the experiment of Figure 4 and add an additional curve (in pink) representing some random or unstructured content. In most cases, this new pink curve is relatively far from the others, supporting the idea that the filtered gradient still contains enough information to allow the model to provide a good decision.
>
> Additionally, we would like to emphasize that the frequency cut-off used in FORGrad does not always correspond to a sigma=10 as done in Figure 4 to minimize the estimation error.
>
> However,  we do understand the reviewer’s concern about the maximum error and we will add a table in section C of the supplementary material containing for each model, the maximal error but also the error associated with the sigma value obtained with FORGrad.
>
> **- The conclusion is a bit off:** Thank you for this suggestion; we agree with you that this sentence in the conclusion was hasty, and we have now toned it down.
>
> **- Why not change the model?** While acknowledging the validity of the reviewer's insights regarding the presence of noisy content in the gradient, we firmly believe that these observations have the potential to enhance current models if a remedy to this phenomenon is found. It's important to note, however, that the core emphasis of this work lies in the realm of explainability. Through our method, we propose to rectify the identified phenomenon to provide significantly improved explanations.
>
> We now answer your question: *How would you describe the relationship of your work to Balduzzi et al. 2017?*: As our work shows that most of the noise is introduced by downsampling operations, we are aware that several operations inside a block can also have an effect on the gradient. (Balduzzi et al., 2017) investigate the gradients of models with vs. without skip connections compared to feedforward models (models without skip connections). The authors additionally provide more detailed evidence of the effect of batch normalization and initialization on the content of the gradients. This paper is therefore complementary to our observations on the noise introduced by specific operations.

---

> > ### Comment · Reviewer_mo1q · 2023-08-16
> > **Answer to the Rebuttal**
> >
> > Dear Authors, I thank you for your rebuttal and I am glad that you addressed some issues like my concerns about the conclusion. However, I would have expected that to see some hard number for my question "How relevant are high-frequencies?", after all the irrelevance of "noise" is cited as one main motivation. Furthermore, I agree with the concerns about the theoretical section form reviewer vA9f and bkGG. Therefore, I downgraded my rating to "borderline reject"

---

> > > ### Author Response · Authors · 2023-08-21
> > >
> > > Thank you for this comment. However, we are surprised by these comments. As requested we have added all the maximum error between the curves in the supplementary material. Additionally, as we mentioned in the general comment of our rebuttal, we worked on the theoretical section and feel that we have addressed all the concerns of reviewers vA9f and bkGG.

---

### Author Rebuttal · Authors · 2023-08-09

We express our heartfelt appreciation to the reviewers for devoting their time to meticulously read and evaluate our paper. Their critiques have been perceptive and thought-provoking. We believe that we were able to address most of their comment in a satisfactory manner and the resulting manuscript has greatly improved. We will begin by addressing common general remarks from multiple reviewers and then proceed to respond to each of their individual comments accordingly.

**- The results rely on metrics that could be biased or not representative enough of the quality of the explanation and we would need a user study to prove our point (mo1q, bkGG).**

We noted that reviewers mo1q and bkGG found that the metrics we used are not sufficient to show the effect of FORGrad on the improvement of attribution methods; they suggested conducting human experiments to strengthen our results. We would like to emphasize that the metrics we used are standard in the field contrary to human psychophysics studies. Most importantly, as these attribution methods were previously evaluated using these same metrics (and not with human experiments), it only makes sense for us to use these same metrics to demonstrate any improvement.

**- We consider ONLY downsampling operations but no other intra-block mechanism as a source of noise (mo1q, vA9f, QucS), minimizing the impact of other operations.**

To be clear: the main focus of this work is to provide a way to improve attribution methods – not investigating in detail all the operations and possible sources of gradient noise. Our limited analyses are only meant to illustrate possible causes for this noise and they are not meant to be exhaustive.

With that said, we ran an additional experiment to help address this criticism. In Figure 1 of the rebuttal, we plot the gradient complexity (measured as Laplacian and Kolmogorov) of different VGG16 layers for different stride lengths. Without any stride (stride = 1) the complexity scores remain relatively constant with depth – unlike with strides. This confirms that strided operations introduce more noise than other operations/mechanisms. One can also gather from the figure that smaller strides lead to a greater increase in complexity. This is consistent with the checkerboard patterns shown in Figure 5. Since a stride of 2 is most commonly used, most modern deep net architectures are thus subject to this potentially problematic observation, justifying the use of FORGrad to counterbalance this artifact to get better explanations.

Similarly, we studied the impact of skip connections on gradients’ complexity and found a small increase in complexity due to this specific mechanism; however, this was not significant compared to strided operations. The resulting figure is now added to the supplementary material.

**- The theoretical section requires more work (vA9f, bkGG).**

We acknowledge that our theoretical (sub)section could have been better tied to the rest of the manuscript. We have therefore thoroughly rewritten it, in order to highlight the theoretical contributions, their relations to the FORGrad and the choice of optimal sigma, and empirical results. In the process, we were able to improve the constants in one of our results.
We answer the specific concerns of reviewers vA9f and bkGG with more details in their respective sections.

**- Sanity checks for reviewer bkGG**

As suggested by reviewer bkGG, we added an additional experiment reproducing the test from Adebayo et al (2018) on a ResNet50. We show in Figure 2 of the rebuttal the results of this Sanity check after applying FORGrad. As expected, because we do not modify any of the methods directly (we don’t change the way they produce the explanation), the methods which failed to pass the test originally (such as Guided BackProp, and DeconvNet) still appear to be sensitive to randomization. Conversely, saliency-based methods remain robust, as shown by Adebayo et al, and are not altered or affected by FORGrad.

---

### Decision · Program_Chairs · 2023-09-21

**Decision:**

Reject

**Comment:**

The paper has 3 borderline rating and one reject. All reviewers considered the rebuttal provided by the authors.

On the positive side, there is the simplicity of the proposed approach and the improvements it brings for some metrics of explanation qualities. On the other hand, reviewers have expressed a diverse and important set of concerns which altogether makes a rather confident case for rejection. These include the various aspects of the formal presentation (e.g., source of the high frequency), the existence of other (sometimes not cited) works in the literature discussing noise in the attribution and suggesting filtering (of frequency), and quite a few strong unsubstantiated statements. Therefore, the paper needs a major revision in its formal presentation of the general statements and specific contributions as well as in providing theoretical and/or conclusive empirical evidence for them.